# A monofluoride ether-based electrolyte solution for fast-charging and low-temperature non-aqueous lithium metal batteries

Guangzhao Zhang[1,7], Jian Chang [1,7], Liguang Wang[2], Jiawei Li[3], Chaoyang Wang [4], Ruo Wang[1], Guoli Shi[1], Kai Yu[1], Wei Huang[5], Honghe Zheng[6], Tianpin Wu [2] ✉, Yonghong Deng [1] ✉ & Jun Lu [2] ✉

The electrochemical stability window of the electrolyte solution limits the energy content of non-aqueous lithium metal batteries. In particular, although electrolytes comprising fluorinated solvents show good oxidation stability against high-voltage positive electrode active materials such as $LiNi_{0.8}Co_{0.1}Mn_{0.1}O_2$ (NCM811), the ionic conductivity is adversely affected and, thus, the battery cycling performance at high current rates and low temperatures. To address these issues, here we report the design and synthesis of a monofluoride ether as an electrolyte solvent with Li-F and Li-O tridentate coordination chemistries. The monofluoro substituent ($-CH_2F$) in the solvent molecule, differently from the difluoro ($-CHF_2$) and trifluoro ($-CF_3$) counterparts, improves the electrolyte ionic conductivity without narrowing the oxidation stability. Indeed, the electrolyte solution with the monofluoride ether solvent demonstrates good compatibility with positive and negative electrodes in a wide range of temperatures (i.e., from $-60\,°C$ to $+60\,°C$) and at high charge/discharge rates (e.g., at $17.5\,mA\,cm^{-2}$). Using this electrolyte solution, we assemble and test a 320 mAh Li||NCM811 multi-layer pouch cell, which delivers a specific energy of $426\,Wh\,kg^{-1}$ (based on the weight of the entire cell) and capacity retention of 80% after 200 cycles at $0.8/8\,mA\,cm^{-2}$ charge/discharge rate and $30\,°C$.

The ever-increasing demand for high-energy-density batteries has motivated revisiting lithium (Li) metal anodes due to the lowest electrochemical redox potential ($-3.04\,V$ vs. standard hydrogen electrode) and high theoretical specific capacity ($3860\,mAh\,g^{-1}$)[1,2]. However, the standard non-aqueous electrolyte solutions in Li-ion batteries fail to achieve stable operation in high-voltage Li metal batteries (LMBs) due to their poor stability against both Li metal anodes and high-voltage cathodes[3,4]. In particular, these instabilities (such as dendritic

[1]Department of Materials Science & Engineering, School of Innovation and Entrepreneurship, Southern University of Science and Technology, Southern University of Science and Technology of China, Shenzhen 518055, China. [2]College of Chemical and Biological Engineering, Zhejiang University, Hangzhou 310027, China. [3]School of Materials Science and Engineering, China University of Petroleum (East China) Qingdao, Qingdao 266580, China. [4]Research Institute of Materials Science, South China University of Technology, Guangzhou 510640, China. [5]National Center for Applied Mathematics Shenzhen (NCAMS, Digital Economy Research Center-DeFin) and College of Business, Southern University of Science and Technology, Shenzhen 518055, China. [6]College of Energy & Collaborative Innovation Center of Suzhou Nano Science and Technology, Soochow University, Suzhou, Jiangsu 215006, China. [7]These authors contributed equally: Guangzhao Zhang, Jian Chang. ✉e-mail: tianpinwu@zju.edu.cn; yhdeng08@163.com; junzoelu@zju.edu.cn

deposition and electrolyte decomposition) will be further amplified under fast-charging and low-temperature conditions[5,6]. The ideal electrolyte for fast-charging and low-temperature LMBs under practical conditions should simultaneously meet several critical requirements, including high stability against electrodes, high ionic conductivity over wide-temperature ranges, low density, and high boiling point[7,8]. Therefore, the successful deployment of such high-voltage LMBs is facing a dilemma in selecting electrolyte systems for fast-charging and wide-temperature applications.

In this regard, several engineering strategies, including high-concentration electrolytes, localized high-concentration electrolytes, additive-regulated electrolytes, and fluorinated electrolytes have been proposed for high-voltage LMBs[9–13]. Among various options, solvent fluorination could enhance the oxidative stability of non-aqueous electrolyte solutions from the molecular perspective by decreasing the highest occupied molecular orbital (HOMO) level of the solvent molecules[14]. Despite the significant development of fluorinated carbonate-based electrolytes, their ion-conducting carbonyl groups are easily reduced by Li metal. Instead, fluorinated ether-based electrolytes are designed to stabilize high-voltage LMBs due to their good compatibility with Li metal and cycling efficiency. In particular, a family of trifluoro- and difluoro-substituted ether-based electrolytes are highly stable with high-voltage cathodes due to their strong electron-withdrawing effects of fluorinated substitution groups[11,13]. However, the ionic conductivity of all these electrolytes is largely reduced after the fluorination, leading to the performance degradation of the battery at high-rate or low-temperature conditions. This is because, on the one hand, difluoro and trifluoro groups largely reduce the electron cloud density of ion-conducting groups in solvent molecules and weaken their dissociation ability with lithium salts. On the other hand, these fluorinated groups themselves usually show weak and even no coordination interactions with Li[+] cations and further induce large ion aggregation and sluggish ion transport. Therefore, it is required to design fluorinated electrolytes with high ionic conductivity, Li metal cyclability, and oxidative stability for practical fast-charging and low-temperature LMBs.

In this work, we design and synthesize monofluoride bis(2-fluoroethyl) ethers (BFE) as electrolyte solvents for fast-charging and low-temperature LMBs by strong Li-F and Li-O tridentate coordination chemistries. By tailoring the fluorination degree, monofluoro substituent (-CH$_2$F) maximizes the ion conductivity of fluorinated electrolytes while retaining high oxidation stability compared to difluoro (-CHF$_2$) and trifluoro (-CF$_3$) counterparts. The single-salt and single-solvent monofluoride electrolytes enable high bulk ionic conductivity (8 mS cm$^{-1}$, 30 °C), Li metal cycling efficiency (99.75%, Li||Cu cell @ 0.5 mA cm$^{-2}$ and 1 mAh cm$^{-2}$), and oxidation stability (4.7 V vs. Li/Li[+]) at wide temperature conditions (−60 °C to +60 °C). Consequently, the monofluoride electrolyte achieves high rate capability (17.5 mA cm$^{-2}$) and stable low-temperature (−30 °C) operation during 100 cycles in 50-μm-thin-Li||high-loading-NCM811 full cells under practical conditions (areal capacity: 3.5 mAh cm$^{-2}$, high cut-off voltage: 4.4 V, low negative-to-positive (N/P) ratio: 2.8, and lean electrolyte: 2.4 g Ah$^{-1}$). At high discharge rates, the practical pouch cell also offers high specific energy (426 Wh kg$^{-1}$) based on the weight of the entire cell and high-capacity retention (>80%) during 200 cycles at 30 °C. The mono-fluoride design and tridentate coordination chemistry offer a feasible pathway toward developing fluorinated electrolytes for industrial high-voltage lithium batteries under extreme conditions (including high loading, high rate, and low temperature).

## Design principle and physicochemical properties of mono-fluoride electrolyte solution

According to the classical Pauling scale[15], fluorinated substitution groups could transfer localized electrons on neighboring polar groups and equally share them due to the strong electron-

withdrawing capability of fluorine atoms. As such, the mono-fluoride (-CH$_2$F) substituted group is expected to have much stronger coordination with Li[+] cations by occupying more localized electrons compared to trifluoro (-CF$_3$) and difluoro (-CHF$_2$) counterparts (Fig. 1a). Based on the above principle and hypothesis, we designed and synthesized monofluoride substituted bis(2-fluoroethyl) ethers as electrolyte solvents to maximize the coordination interaction between Li[+] cations and the solvent by forming strong Li-F and Li-O tridentate coordination interactions (Fig. 1b). Here, two methylene groups are designed between the monofluoride and oxygen groups to weaken their intramolecular interactions and promote the stable five-member-ring coordination with Li[+] ions. The target BFE molecules can be synthesized by a one-step nucleophilic substitution reaction under mild conditions with only two basic low-cost reagents (Fig. 1c). Accordingly, the monofluoride electrolyte combining single-salt (lithium bis(fluorosulfonyl)imide, LiFSI) and single-solvent (BFE) is expected to simultaneously achieve high ionic conductivity and oxidation stability together with good Li metal cycling efficiency (Supplementary Fig. 1), which is found to outperform their counterparts of non-fluorinated diethyl ether (DEE) and fully fluorinated bis(2,2,2-trifluoroethyl) ether (BTFE). Due to the low freezing point, DEE has been recognized as a low-temperature electrolyte solvent but suffers from severe oxidation instability, low boiling point, and relatively low ionic conductivity in LMBs[16]. After terminally functionalizing the DEE with -CF$_3$ groups, the oxidation stability of BTFE is enhanced[17]. However, the incorporation of BTFE reduces the overall ionic conductivity of fluorinated electrolytes due to the inert interaction between Li[+] ions and the BTFE solvent.

The successful synthesis of the BFE is confirmed via proton and fluorine nuclear magnetic resonance spectroscopy (NMR) in Fig. 1d. We found that the BFE solvent exhibits a lower density (Fig. 1e, 0.98 g cm$^{-3}$) compared to commercial carbonate solvents (>1.0 g cm$^{-3}$) and previously reported fluorinated solvents (>1.3 g cm$^{-3}$) at 30 °C[12,13,18–20], which is expected to improve the energy density of the whole battery. Moreover, the BFE solvent is found to have the highest boiling point (128 °C) among non-fluorinated (DEE, 34.5 °C), partially and fully fluorinated (BTFE, 63 °C) solvents (Supplementary Table 1), which is ascribed to the strong F···H hydrogen bonding interactions among BFE molecules. The high boiling point of the BFE solvent ensures that it can maintain high ionic conductivity under wide-temperature conditions. In addition, the introduction of monofluoride groups could also improve the safety of the whole battery by reducing the flammability of the monofluoride electrolyte (Supplementary Fig. 2).

To reveal the impact of monofluoride substitution on the ionic conductivity and oxidative stability of fluorinated non-aqueous electrolytes, a typical LiFSI salt is selected to dissolve into the BFE solvent to form a single-salt and single-solvent electrolyte. It is found that the viscosity of the monofluoride BFE electrolyte is slightly higher than that of the non-fluorinated DEE counterpart (Fig. 1f, measured at 30 °C). However, the BFE electrolyte shows much higher ionic conductivities within various salt concentrations ranging from 0.5 to 2.5 M compared to non-fluorinated DEE. For example, a 2 M LiFSI/BFE electrolyte exhibits the highest ionic conductivity of 8 mS cm$^{-1}$ among various salt concentrations at 30 °C. Moreover, the BFE electrolyte also maintains high ionic conductivities (0.95 ~ 15 mS cm$^{-1}$) and high Li[+] transference numbers (0.69) at a wide range of temperatures ranging from −60 to 70 °C (Fig. 1g and Supplementary Fig. 3), which is well contrasted with non-fluorinated DEE electrolyte. Importantly, linear sweep voltammetry (LSV) of Li||Al cell indicates BFE electrolyte could provide an electrochemical stability window of 4.7 V (vs. Li[+]/Li) compared to 1,2-dimethoxyethane (DME, an ether solvent commonly used for LMBs, 3.8 V) and DEE (4.4 V) (Supplementary Fig. 4a). A much larger oxidation potential of 5.2 V is obtained in the testing Li||Pt cell after

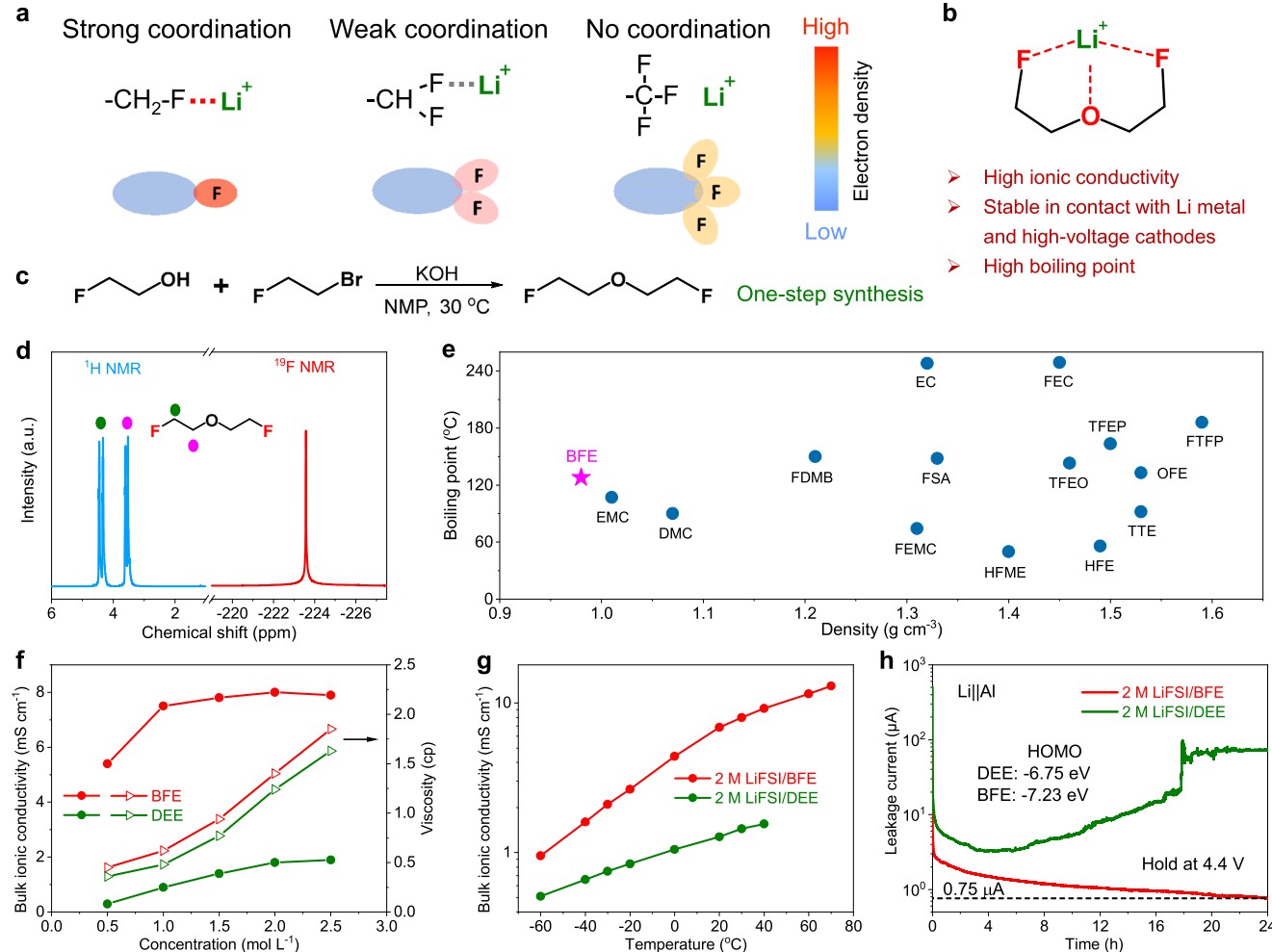

**Fig. 1 | Physicochemical and electrochemical characterizations of the mono-fluoride ether-based electrolyte with tridentate coordination chemistry.** **a** Coordination chemistry of monofluoride, difluoro, and trifluoro groups. **b** The molecular design of the BFE, can coordinate $Li^+$ ions with one Li-O and two Li-F interactions simultaneously. **c** One-step synthesis of BFE solvent at 30 °C. **d** $^1H$, and $^{19}F$ NMR spectra of BFE molecule. **e** Boiling point and density of BFE and conventional fluorinated/non-fluorinated solvents at 30 °C. **f** Bulk ionic conductivities, and viscosities of BFE and DEE electrolytes with various salt concentrations at 30 °C. **g** Measured bulk ionic conductivity of BFE and DEE electrolytes as a function of temperatures. **h** Leakage currents of Li||Al cells using BFE and DEE electrolytes at a constant applied voltage of 4.4 V and a temperature of 30 °C. Insert shows the HOMO levels for DEE and BFE, respectively.

using the relatively stable Pt counter electrode (Supplementary Fig. 4b). In addition, potentiostatic polarization measurements of Li||Al cells with various electrolytes were also performed to detect the oxidation leakage current. The Li||Al cell with BFE electrolyte shows the smallest leakage current under various applied voltages from 4.0 to 4.8 V, while large leakage currents are observed for DME and DEE electrolytes at about 4.0 and 4.4 V, respectively (Supplementary Fig. 4c). For example, the leakage current of the cell using BFE electrolyte shows a gradual decrease trend when a constant voltage of 4.4 V is applied, and then reaches a minimum value of 0.75 μA cm⁻² after holding 24 h, which is two orders of magnitude lower than DEE-based cell (Fig. 1h). The oxidative stability improvement of BFE electrolyte should be ascribed to the lower HOMO value of −7.23 eV compared to DEE (−6.75 eV). The ex situ scanning electron microscopy (SEM) images of Al foils are also examined after holding the Li||Al cells at 4.8 V for 5 h with various electrolytes. The extracted Al foil after polarization treatment shows a smooth morphology in the BFE electrolyte, while corrosion and cracks are observed for Al foils cycled in DME and DEE electrolytes, respectively (Supplementary Fig. 5). In addition, the ex situ XPS measurements and analyses also reveals corrosion-resistant $LiF/AlF_3$ components are formed at the surface of Al foils in the BFE electrolyte (Supplementary Fig. 6).

## Boosting the electrolyte ionic conductivity via monofluoride substitution in the solvent

To understand the mechanism behind the high ionic conductivity of monofluoride-based electrolytes, we conducted density functional theory (DFT) calculations. In principle, the ionic conductivity of electrolytes mainly depends on the coordination interaction between $Li^+$ cations and the solvent molecules. Here, the electrostatic potentials of fluorinated solvent molecules with different fluorination degrees, including DEE, BTFE, bis(difluoroethyl) ether (BDE), 2,2,2-trifluoroethyl 2-fluoroethyl ether (TFFE), 1,1-difluoroethyl-2-fluoroethyl ether (DFE) and BFE, were investigated to reveal the electron density distribution of the whole molecule (Fig. 2a). Compared to the non-fluorinated DEE, both BTFE and BDE molecules exhibit low electron densities near the middle oxygen atom after the trifluoro and difluoro substitution, respectively. These multifluoride substitution groups withdraw the localized electrons from the middle oxygen atoms, leading to poor coordination with $Li^+$ cations. As the degree of local fluorination decreases, the electron density of the middle oxygen atom is gradually restored in both TFFE and DFE. In particular, the monofluoride atoms in both TFFE and DFE also show comparable electron densities with the middle oxygen atom. It is worth noting that the difluoro group in DFE shows slightly higher electron densities than the trifluoro group of

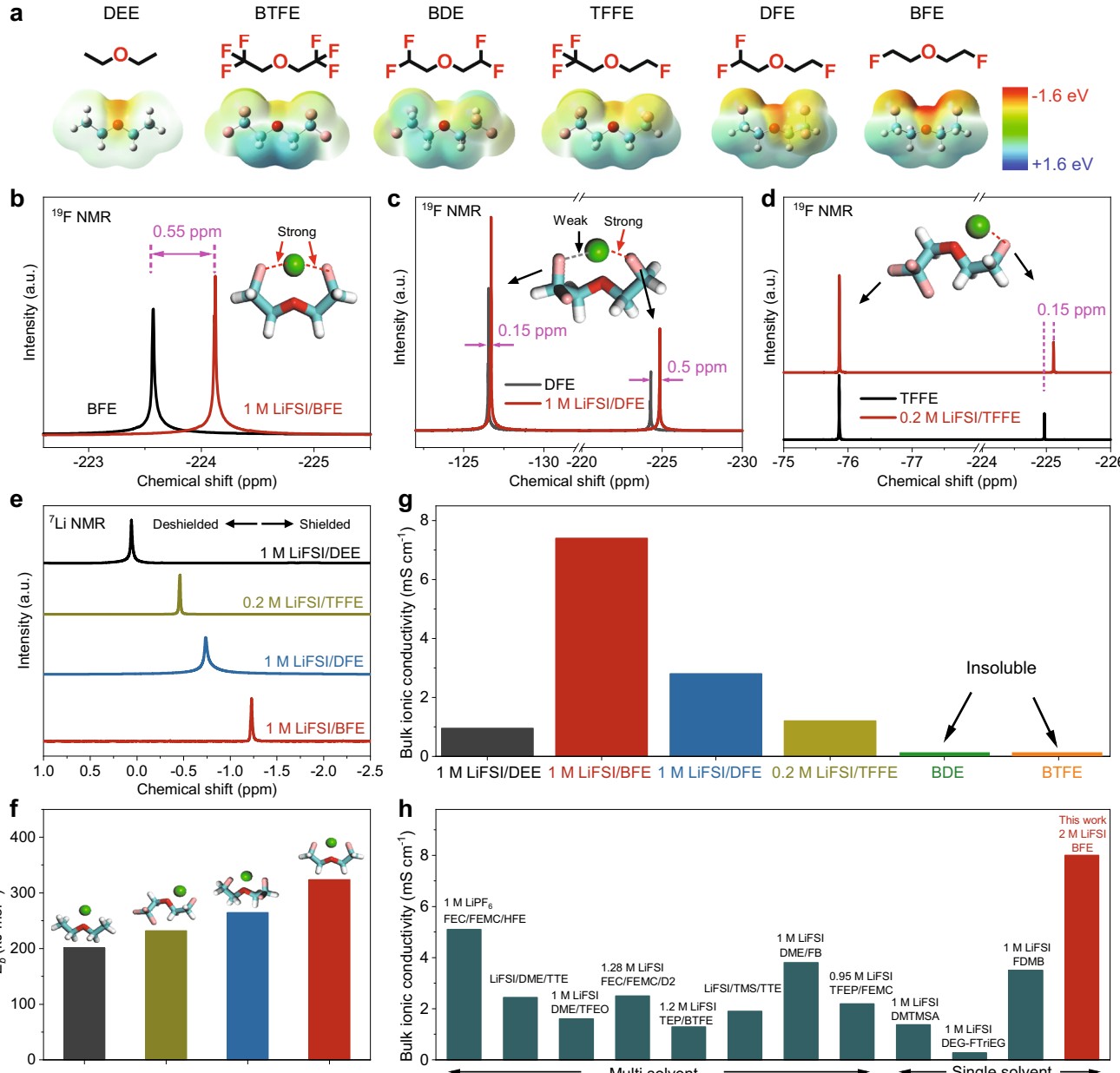

**Fig. 2 | Structure-property relationships between fluorination degree and ionic conductivity. a** Comparison of chemical structures and electrostatic potentials (ESP) of various ether solvents including DEE, BTFE, BDE, TFFE, DFE, and BFE. **b–d** $^{19}$F NMR of BFE, DFE, and TFFE before and after the salt dissolution (Reference: CF$_3$COOH). Green sphere represents Li$^+$ ion, while sticks in white, pink, light blue and red color represent H, F, C, and O atoms, respectively. **e** $^7$Li NMR of LiFSI salts in different electrolytes. **f** Solvating energies of DEE, TFFE, DFE, and BFE electrolytes. **g** Bulk ionic conductivities of various electrolytes at 30 °C, including DEE, BTFE, BDE, TFFE, DFE, and BFE. **h** Bulk ionic conductivity of BFE and previously reported fluorinated electrolytes at 30 °C. Because LiFSI is almost insoluble in BTFE and BDE, the bulk ionic conductivity value is not reported in the chart.

TFFE, suggesting difluoro groups have relatively higher coordination strength with Li$^+$ cations. Interestingly, the monofluoride BFE exhibits the highest electron densities around both the middle oxygen atom and terminal single fluorine atom among all the fluorinated molecules. Consequently, reducing the degree of fluorination can gradually stimulate the coordination ability of fluorine atoms themselves with Li$^+$ cations in fluorinated solvents. Due to the electron-withdrawing effect of fluorinated groups, the electron density from oxygen in BFE is indeed lower than that of DEE (BFE-1) in the linear molecular configuration (Supplementary Fig. 7). However, the linear configuration of the BFE molecule can be transformed into a stable five-member-ring molecular structure after coordinating with Li$^+$ ions (BFE-2). The electron density of oxygen in BFE can be significantly enhanced by sharing

the delocalized electrons of monofluoride atoms. This phenomenon also indirectly confirms the interaction of Li-F in BFE electrolytes.

Under the guidance of theoretical calculation, fluorine NMR measurements are carried out to distinguish the coordination interactions of monofluoride, difluoro, and trifluoro substitution groups with Li$^+$ cations by using the above-fluorinated solvents. As shown in Fig. 2b, a large upfield shift of 0.55 ppm is observed for the BFE electrolyte after dissolving 1 M LiFSI, indicating the strong coordination interactions between the monofluoride groups and Li$^+$ cations. The shielded effect of fluorine nuclei is amplified by increasing the salt concentration (Supplementary Fig. 8). Also, the strong coordination interaction between the middle oxygen group and Li$^+$ cations is revealed in the BFE electrolyte, as confirmed by $^{17}$O NMR

(Supplementary Fig. 9) and the Raman spectra (Supplementary Fig. 10). In addition, two asymmetric monofluoride molecules of DFE and TFEE were also synthesized for comparison (Supplementary Figs. 11 and 12). The asymmetric DFE molecule exhibits two characteristic peaks, corresponding to the monofluoride and difluoro-substitution groups, respectively (Fig. 2c). After the dissolution of the LiFSI (1 M), upfield shifts are shown in both the representative peaks of the monofluoride (0.5 ppm) and difluoro-substitution (0.15 ppm) groups. Compared to difluoro counterparts, the larger chemical shift change of the monofluoride peak indicates a stronger coordination interaction with Li+ cations. In comparison, the TFFE molecule with monofluoride and trifluoro terminal groups could barely provide a maximum salt solubility of 0.2 M due to the insufficient solvation ability. Even at 0.2 M, the monofluoride peak of the TFFE electrolyte still exhibits an upfield shift (0.15 ppm) while the trifluoro peak is hardly shifted, indicating almost no coordination interactions with Li+ cations (Fig. 2d).

The strong coordination interaction of monofluoride groups could enable fast ionic transport behaviors of monofluoride electrolytes by regulating the solvation capability. Here, $^7$Li NMR characterization and theoretical calculation are conducted to analyze the solvation capability and solvent-Li+ interactions of DEE, TFFE, DFE, and BFE. In $^7$Li NMR spectra, a lower chemical shift represents a more shielded effect of Li nuclei due to the strong coordination of surrounding ligands. As shown in Fig. 2e, the lowest chemical shift is observed for BFE electrolytes (−1.24 ppm), indicating a stronger solvation capability than DFE (−0.75 ppm), TFFE (−0.5 ppm), and DEE (0 ppm). The solvation trend of various fluorinated solvents is well-accorded with the calculated solvent-Li+ solvating energy in the order of DEE (200 kJ mol$^{-1}$) < TFFE (225 kJ mol$^{-1}$) < DFE (250 kJ mol$^{-1}$) < BFE (325 kJ mol$^{-1}$) (Fig. 2f). This result is also supported by the chemical shift change of $^{19}$F in FSI− anions (Supplementary Fig. 13). Accordingly, the ionic conductivity of fluorinated electrolytes gradually increases in the order of DEE < TFFE < DFE < BFE (Fig. 2g). It is worth noting that both highly fluorinated BTFE and BDE molecules are difficult to dissolve lithium salts due to their low solvation capability. As a result, our single-salt and single-solvent monofluoride electrolyte exhibit the highest ionic conductivity among current state-of-the-art single-solvent and complex fluorinated electrolytes (Fig. 2h, at 30 °C)[12,18,19,21–26].

## Understanding the solvation structure of the monofluoride ether-based electrolyte

The ion transport behavior of electrolytes is determined by the solvation structure within the complexation of cations with solvents and anions. Here, molecular dynamics (MD) simulations are performed to investigate the solvation structure and the Li+-solvate distribution of monofluoride electrolytes. Due to the low solvating power of DEE molecules, lithium salts cannot be fully dissociated in DEE electrolyte, leading to large and unevenly distributed solvation clusters containing a large number of FSI− anions at 30 °C (Fig. 3a). These relatively large solvation clusters are difficult to move under an electric field, resulting in the sluggish transport of solvated Li+ cations. As shown in Fig. 3b, much larger solvation clusters were observed when the temperature decreased to −30 °C. Unlike DEE, the snapshot of the BFE electrolytes reveals large amounts of small and well-distributed solvation clusters at 30 °C (Fig. 3c), which is attributed to the higher solvation power of BFE to Li+ cations (Fig. 2f). Importantly, the homogeneous morphology is maintained even when the temperature is reduced to −30 °C (Fig. 3d). Therefore, the fast ion transport behavior of monofluoride electrolytes mainly contributes to the formation of small-sized and well-distributed solvation clusters at both 30 and −30 °C.

To reveal the precise composition of the solvation structures, we then analyzed the radical distribution functions (RDF) of the DEE and BFE electrolytes. For the DEE, peaks of Li-O$_{FSI}$ and Li-O$_{DEE}$ are observed

around the distance of 2 Å at 30 °C (Fig. 3e). However, the intensity of Li-O$_{FSI}$ is roughly an order of magnitude higher than that of Li-O$_{DEE}$, suggesting that the primary solvation shell is mainly composed of non-fully dissociated LiFSI with a minor amount of DEE solvent. After lowering the temperature to −30 °C, the DEE molecules are partially excluded from the primary solvation shell, indicating the reduced solvating ability of DEE molecules at low temperatures. In comparison, the RDF of BFE electrolyte shows a higher intensity of Li-O$_{BFE}$ peak than that of Li-O$_{FSI}$ at the distance of 2 Å, suggesting most LiFSI are fully dissociated by BFE molecules. Moreover, the appearing Li-F$_{BFE}$ peak around the distance of 2 Å indicates the strong Li-F interactions, facilitating the formation of Li-F and Li-O tridentate coordination interaction. The solvation clusters in the BFE electrolyte were extracted from the MD simulation trajectories. As shown in Supplementary Fig. 14, the clusters of two BFE molecules coordinating one Li+ ion with strong Li-F and Li-O interactions are observed, which belong to the typical solvent-separated ion pair (SSIP) structure and represent 40% of the total lithium in the electrolyte solution (calculated according to the coordination number, Supplementary Fig. 15). This SSIP structure is also confirmed by electrospray ionization mass spectrometry (ESI-MS) with a strong [solvent-Li+] mass peak at the mass of 117 in Supplementary Fig. 16. The other clusters consisting of one Li+ ion, two BFE molecules, and one FSI− anion are also observed in BFE electrolyte (Supplementary Fig. 14). The solvation structure belongs to the contact ion pair (CIP) structure and occupies about 40% of total Li+ ions. An interesting phenomenon is that once FSI− anion participates in the coordination with Li+ ions, a fluorine group of BFE molecule is excluded from the Li+ ion center, keeping the total coordination number of 6. This phenomenon indicates that the coordination strength of Li-F$_{BFE}$ is similar to that of Li-O$_{FSI}$. Interestingly, after decreasing the temperature to −30 °C, the tridentate coordination solvate of Li-F and Li-O is still maintained which is consistent with the morphology results in Fig. 3d. The fast ion transport behavior of monofluoride electrolyte at both 30 and −30 °C is also well-accorded with high diffusivities of Li+ ions in mean squared displacements (MSDs) calculations (Fig. 3f).

The monofluoride electrolyte represents a peculiar solvation structure of solvent-separated ion pairs SSIPs and CIPs as shown via Raman measurements (Supplementary Fig. 17). The S-N-S bending peak of the FSI− anions in the DEE electrolyte undergoes a minor shift from 774 to 750 cm$^{-1}$ after the salt dissolution, suggesting the strong interaction of Li+/FSI− in CIPs structures. In comparison, the S-N-S bending peak of the FSI− anions is markedly shifted to 722 cm$^{-1}$ after dissolving the salts into the DME solvents, indicating the weak interaction of Li+/FSI− in SSIP structures. This result is consistent with the above MD simulation. The BFE electrolyte exhibits a moderate bending peak (735 cm$^{-1}$) of the FSI− anions between DME and DEE electrolytes, indicative of the existence of SSIP and CIP solvation structures. The SSIPs in the BFE electrolyte could substantially improve the overall ion conductivity by promoting the effective dissociation of the LiFSI salt. On the other hand, the CIPs in BFE electrolytes could stabilize the interface of Li metal anode and high-voltage cathode by decomposing the anions in solvation structures and forming LiF components.

## Electrochemical characterization of the monofluoride ether-based electrolyte in Li metal cells

Apart from ionic conductivity and extended electrochemical stability, high Li metal compatibility of electrolytes is also required for long-term cycling LMBs. Here, the stability of the BFE electrolyte against Li metal anode is estimated by recording the average Coulombic efficiency (CE) of Li metal in Li||Cu asymmetric cells applying the Aurbach method during the partial plating and stripping processes[27]. As disclosed in Fig. 4a, the BFE electrolyte with high ionic conductivity could offer a high Li metal CE up to 99.75% (Supplementary Fig. 18) compared to DEE (99.21%) and DME (98.65%). The BFE electrolyte also exhibits a lower overpotential (5 mV) during the cycling process than

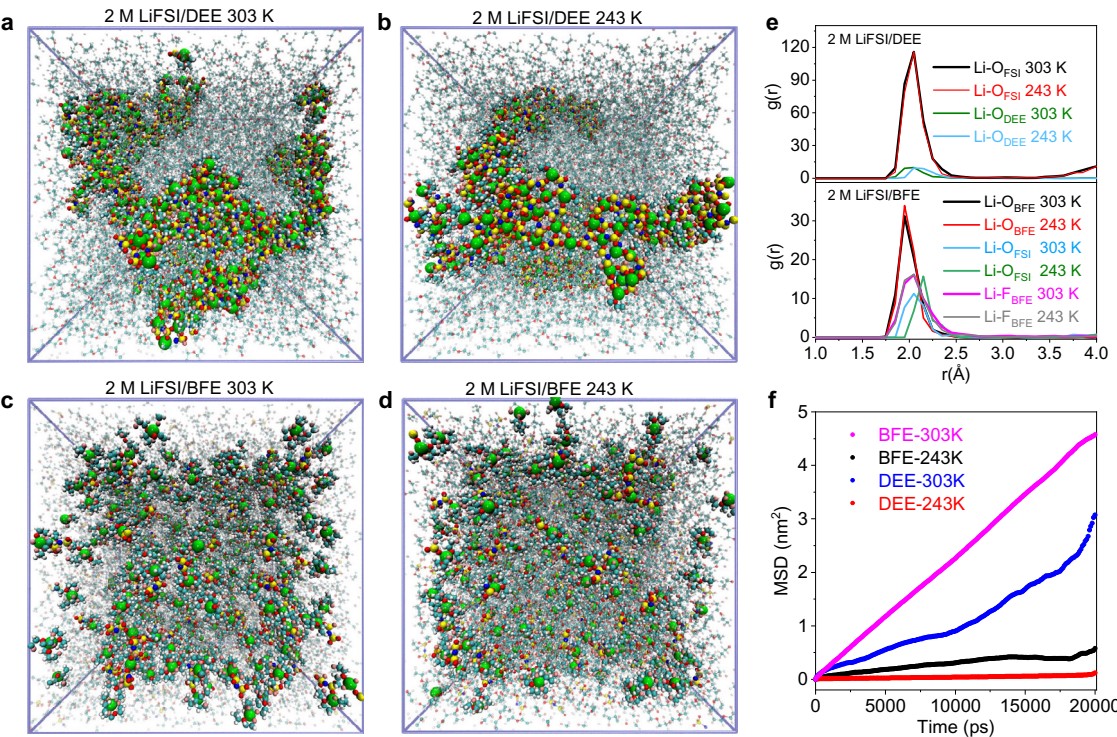

**Fig. 3 | Investigation of solvation structures via molecular dynamic simulations. a–d** Molecular dynamic (MD) simulation trajectories of DEE (**a**, **b**) and BFE (**c**, **d**) electrolytes at −30 and 30 °C, respectively. Li⁺ ions, coordinated solvents, and FSI⁻ anions (within 2.5 Å) are depicted in a ball-and-stick model while the free solvents and FSI⁻ anions are colored semitransparency. Colors for different elements: H-white, Li-lime, C-cyan, N-blue, O-red, F-pink, S-yellow. **e** Radical distribution functions comparison of Li-O (BFE), Li-F (BFE), and Li-O (FSI) pairs for DEE and BEE electrolytes at −30 and 30 °C. **f** MSD of Li⁺ ions in BFE and DEE at −30 and 30 °C.

DEE (15 mV) and DME (29 mV) counterparts (Supplementary Fig. 19). The ability to stabilize Li metal with BFE electrolyte becomes more prominent when the striping/plating process is carried out at high current densities (Supplementary Fig. 20). The assembled symmetric cell of Li‖Li with BFE electrolyte always shows less overpotential than DEE and DME electrolytes at a wide range of current densities ranging from 0.5 to 10 mA cm⁻² (Fig. 4b). At a practical current density of 1.0 mA cm⁻², the interfacial stability of Li metal in various electrolytes (BFE, DME, and DEE) was further evaluated using Li‖Li symmetric cells with a cycling capacity of 1.0 mAh cm⁻². Compared to DME and DEE electrolytes, the symmetric cell in BFE electrolyte shows the smallest overpotential and negligible fluctuation during repeated plating/ stripping processes (Supplementary Fig. 21). In addition, the symmetrical cell with BFE electrolyte also exhibits the smallest charge transfer resistance in various electrolytes, which conforms to the trend of over-potential polarization (Supplementary Fig. 22). Moreover, smooth Li metal deposits are still observed after 250 cycles as shown in SEM images of Li metal electrodes cycled with BFE electrolyte, which is well-contrasted with DME and DEE electrolytes (Supplementary Fig. 23).

The monofluoride ether-based electrolyte was also tested to verify its capability for the development of fast-charging and high-voltage LMBs under practical conditions. Here, high-voltage 50-µm-thin-Li‖ high-loading-NCM811 coin cells with a high areal capacity of 3.5 mAh cm⁻² are assembled with BFE, DME, and DEE electrolytes and cycled at a potential window of 2.8–4.4 V a temperature of 30 °C (Fig. 4c). Benefiting from interfacial compatibility and ionic conductivity of monofluoride electrolyte, the assembled Li‖NCM811 cell delivers specific capacities of 214, 202, 191, 179, 161, and 148 mAh g⁻¹, at 0.7, 1.75, 3.5, 7, 10.5 and 17.5 mA cm⁻², respectively (Fig. 4c, according to the mass of the active material in the positive electrode, Supplementary Fig. 24). The cycling performance we report are well aligned with the state-of-the-art literature of high-voltage LMBs with areal capacities >3 mAh cm⁻² [12,13,22]. In comparison, the Li‖NCM811 cell with DME

electrolyte offers comparable capacities of 208, 200, 186, and 173 mAh g⁻¹ with charge/discharge rates ranging from 0.7 to 7 mA cm⁻² (at 30 °C). However, the cell still suffers rapid capacity degradation once the charge/discharge rate is increased to 10.5 and 17.5 mA cm⁻². Due to the limitation of low ionic conductivity, the cell with DEE electrolyte barely maintains a reversible capacity of 206 mAh g⁻¹ at 0.7 mA cm⁻², while the capacity is drastically reduced to 32 mAh g⁻¹ at 17.5 mA cm⁻² (at 30 °C). Importantly, the cells with BFE electrolytes exhibit excellent cycling stability with high capacities retention of >80% after 300 cycles at both high rates of 3.5 and 7 mA cm⁻², which outperforms DEE and DME electrolytes (at 30 °C, Supplementary Fig. 25).

To examine the practicality of monofluoride electrolytes, the cycling stability of high-voltage Li metal cells is estimated under critical conditions, including high mass loading, limited Li excess, and lean electrolyte. The high-voltage Li‖NCM811 coin cell with an areal capacity of 3.5 mAh cm⁻² and a low N/P ratio of 2.8 is assembled in BFE electrolyte and exhibits good cycling stability with capacities retention of >90% after 200 cycles (at 30 °C, Fig. 4d). In comparison, the Li‖ NCM811 full cells with DEE and DME electrolytes are barely operated for 50 and 75 cycles at a high rate of 7 mA cm⁻² due to the decomposition of electrolyte and detrimental side reactions. At the same capacity of 3.5 mAh cm⁻², the Li‖NCM811 cell with a small N/P ratio of 1.1 can be still operated for 200 cycles with capacities retention of >87% (at 30 °C, Supplementary Fig. 26). In addition, we also assembled the anode-free full cell by using NCM811 as high-voltage positive electrode active material and copper foil as negative electrode. When testing using a constant-current-constant-voltage protocol, a method that is widely used in the practical battery [28–30], the Li‖NCM811 cell also can keep high capacity retention (93%) after 100 cycles (at 1 mA cm⁻² charge/discharge and 30 °C, Supplementary Fig. 27). In addition, the Cu‖NCM811 cell in BFE electrolyte also shows capacity retention of 88% during 40 cycles (30 °C, Supplementary Fig. 28). To maximize the

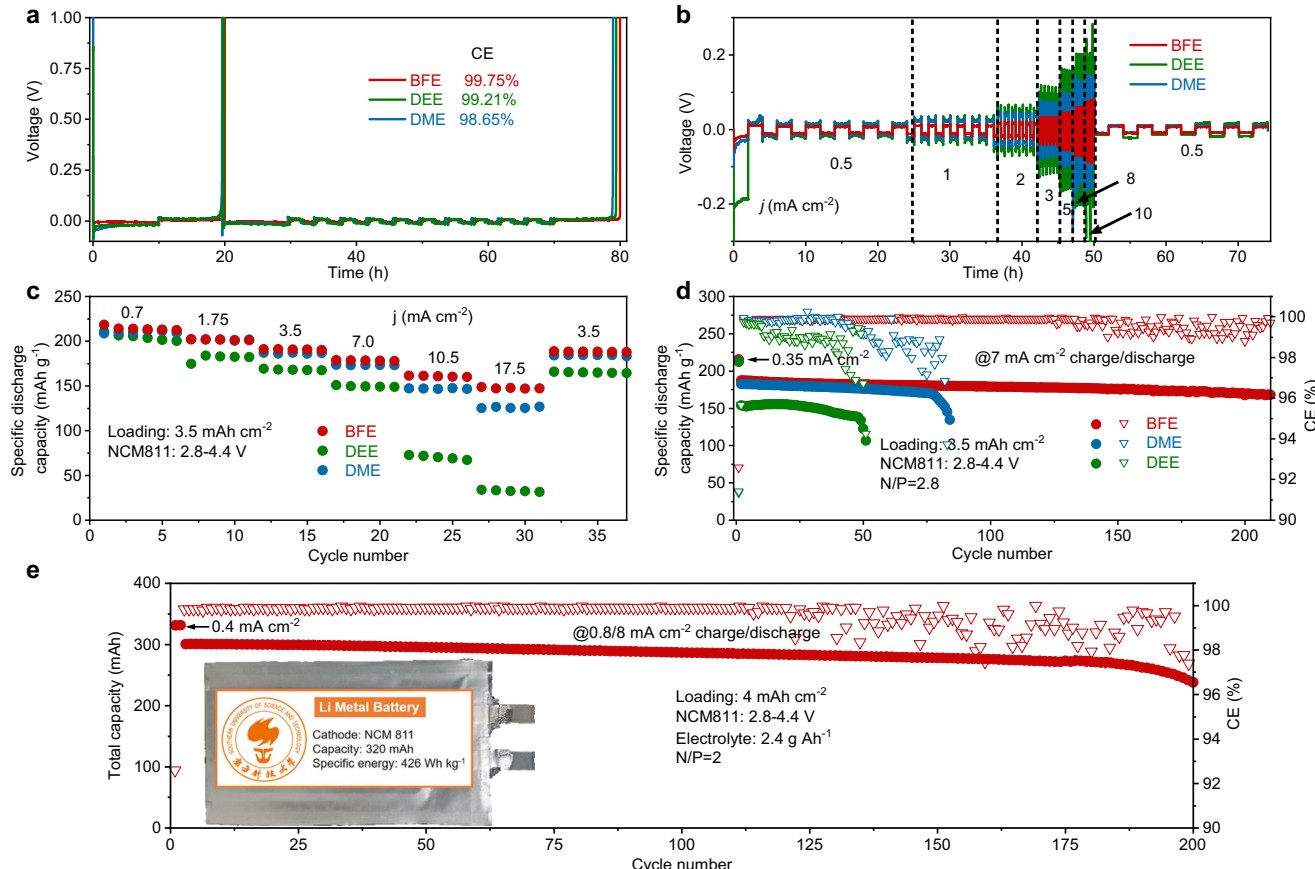

**Fig. 4 | Electrochemical energy storage characterizations of the monofluoride ether-based electrolyte in various Li metal cell configurations. a** Li plating/stripping CEs evaluated via Aurbach's measurement using Li||Cu coin cells at 30 °C. **b** Rate performance of Li||Li symmetric coin cells at current densities from 0.5 to 10 mA cm⁻² in different electrolytes at 30 °C. **c** Rate performance of Li||NCM811 coin cells with BFE, DEE, and DME electrolytes. **d** Cycling performance of Li||NCM811 coin cells with an areal capacity of 3.5 mAh cm⁻², a low N/P ratio of 2.8, and a high current density of 7.0 mA cm⁻² after two formation cycles at 0.35 mA cm⁻². **e** Cycling stability of Li||NCM811 pouch cells (double-sided coated electrodes with four layers of high-voltage positive electrode and five layers of Li metal negative electrode) under practical conditions (areal capacity: 4 mAh cm⁻², N/P: 2, lean electrolyte: 2.4 g Ah⁻¹. Cycling at 0.8/8 mA cm⁻² charge/discharge after two formation cycles at 0.4 mA cm⁻².). Insert is the optical image of the 320-mAh-level pouch cell.

battery specific energy[31], a 320 mAh Li||NCM811 pouch cell is also fabricated under strict conditions (areal capacity: 4 mAh cm⁻², N/P: 2, and electrolyte: 2.4 g Ah⁻¹). The prototype Li metal pouch cell could provide high specific energy of 426 Wh kg⁻¹ (at the current density of 0.4 mA cm⁻²) based on the total weight of the entire cell, including the current collector, electrode, separator, and electrolyte (Fig. 4e and Supplementary Table 2). At charge/discharge current densities of 0.7/7 mA cm⁻², the prototype Li metal pouch cell with lean BFE electrolyte still maintains long cycling stability with high capacities retention of >80% after 200 cycles (tested at 30 °C and calculated from the second cycle). In addition, it is found that the BFE electrolyte is also compatible with the commercial graphite anode in conventional Li-ion batteries. The assembled Li||graphite coin cell with a high cycled capacity of 3.7 mAh cm⁻² shows high capacities retention of >85% during 300 charge/discharge cycles (at 30 °C, 1.85 mA cm⁻², Supplementary Fig. 29).

### Ex situ physicochemical investigations of the Li metal and NCM811-based electrodes

The cycling stability of Li metal batteries with monofluoride electrolytes is also correlated to the interphase components of the Li metal anode and high-voltage NCM811 cathode. Here, 5 mAh cm⁻² of Li metal is electrodeposited onto bare Cu foil to examine the deposition morphology and solid electrolyte interphase (SEI) components of Li metals in various electrolytes, including BFE, DME, and DEE. The scanning

electron microscopy (SEM) image reveals smooth and compact deposit morphologies of Li metal in BFE electrolyte (Fig. 5a). The compact packing and large crystal size of Li deposits are beneficial to the improvement of cycling stability by reducing the formation of thick SEI and dead Li (i.e., Li metal regions which are electronically disconnected from the current collector). However, large amounts of porous Li deposits in both DME and DEE electrolytes can induce continuous SEI and dendritic growth (Fig. 5b, c). At the same plating capacity, the plated Li metal layer in the BFE electrolyte is thinner than those obtained with the DME and DEE electrolytes (Fig. 5d–f). The SEI composition of Li metal anode after 50 plating/stripping cycles in BFE electrolyte was also analyzed by low-dose Cryo transmission electron microscopy (Cryo-TEM) (Fig. 5g–i). A dense and uniform SEI layer with a thickness of 18 nm is observed in Cryo-TEM images. The inverse fast Fourier transform (IFFT) characterization displays that the magnified lattice spacing of 0.20 nm is well-matched with the (200) crystal plane of LiF (Fig. 5h) and the lattice spacing of 0.27 nm is accorded with the (111) crystal planes of Li₂O (Fig. 5i). This result is also confirmed by the corresponding electron energy loss spectroscopy (EELS) measurements (Supplementary Fig. 30). In addition, XPS with etching depth profiles were also used to analyze the SEI components of Li metal in various electrolytes (Fig. 5j). As shown in high-resolution F 1s spectra, the cycled Li metal in BFE electrolyte shows uniform LiF SEI components (-685 eV) in the whole etching depth, while the SEI component in DME and DEE electrolyte gradually decreases. The other peak around

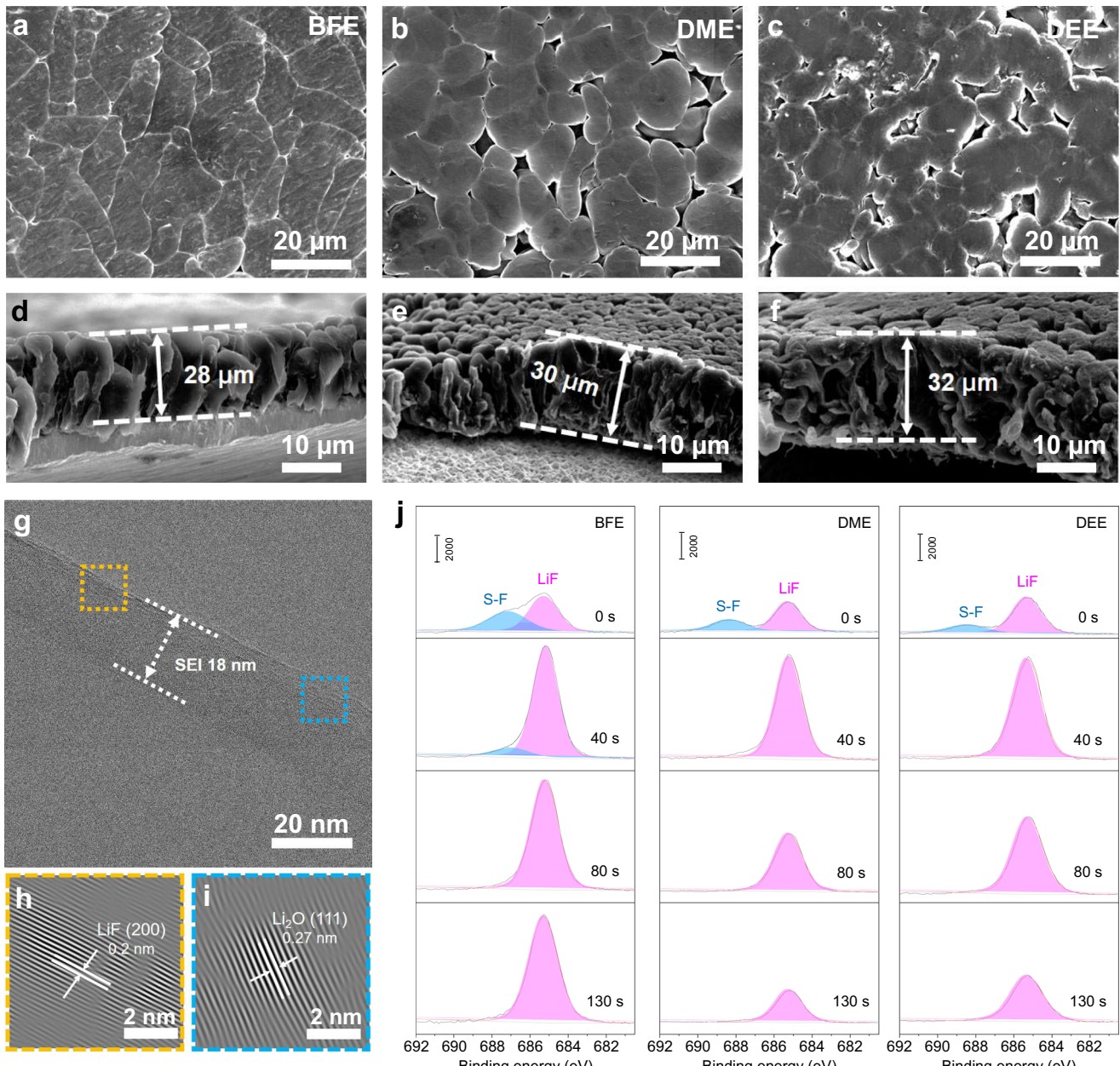

**Fig. 5 | Ex situ interface characterizations of cycled Li metal electrodes. a–f** Top-view and cross-sectional SEM images of plated Li metal on Cu foil after 5 cycles at 0.5 mA cm⁻² with a cutoff capacity of 5 mAh cm⁻² at 30 °C, including BFE (**a, d**), DME (**b, e**), and DEE (**c, f**). **g–i** Cryo-TEM (**g**), and IFFT (**h, i**) images of cycled Li metal on Cu grid in BFE electrolyte (the Li metal was plated on the Cu grid via Li‖Cu cells through electrochemical deposition). **j** F1s XPS spectra with etching depth profiles of the cycled Li metals in different electrolytes (BFE, DME, and DEE). The Li‖Cu cells were cycled at 0.5 mA cm⁻² with a cutoff capacity of 5 mAh cm⁻² at 30 °C for 5 cycles before the XPS measurements. All tests were performed using coin cells.

688 eV is ascribed to S-F components related to the anion decomposition. In addition, the C1s and O1s spectra show that the external SEI of Li metal in BFE electrolyte is mainly organic, while the internal SEI is dominated by inorganic LiF and $Li_2O$ components, which are also accorded with the Cryo-TEM results (Supplementary Figs. 31 and 32). It is worth noting that $LiN_xO_y$ and $Li_2S_x$ are also recognized in S2p and N1s spectra for the cycled Li metal with BFE electrolyte (Supplementary Figs. 33 and 34).

In addition to the stable SEI of the Li metal anode, the monofluoride electrolyte also promotes the formation of LiF-rich cathode electrolyte interphase (CEI) on a high-voltage NCM811 cathode. The CEI components on NCM811 cathodes after 50 charge/discharge cycles are also characterized through high-resolution TEM measurements

and analyses. As observed in Fig. 6a, a uniform CEI layer with a thickness of <1 nm is formed on the surface of the cycled cathode in the BFE electrolyte. Due to the excessive decomposition of the solvents and anions, thicker CEI layers are generated onto the NCM811 cathode in DME and DEE electrolytes, respectively (Fig. 6b, c). In addition, the XPS characterization was used to further analyze the CEI components of cycled cathodes in various electrolytes. As shown in Fig. 6d, e, the high-resolution C 1 s and F 1 s spectra formed in the cycled cathode indicate the CEI of BFE electrolyte has more LiF inorganics and fewer organic components than those obtained with DME and DEE electrolytes. This result indicates the CEI components of the cycled cathode in the BFE electrolyte primarily originate from the anion decomposition, which is also confirmed by the S2p and N1s analysis (Supplementary Fig. 35).

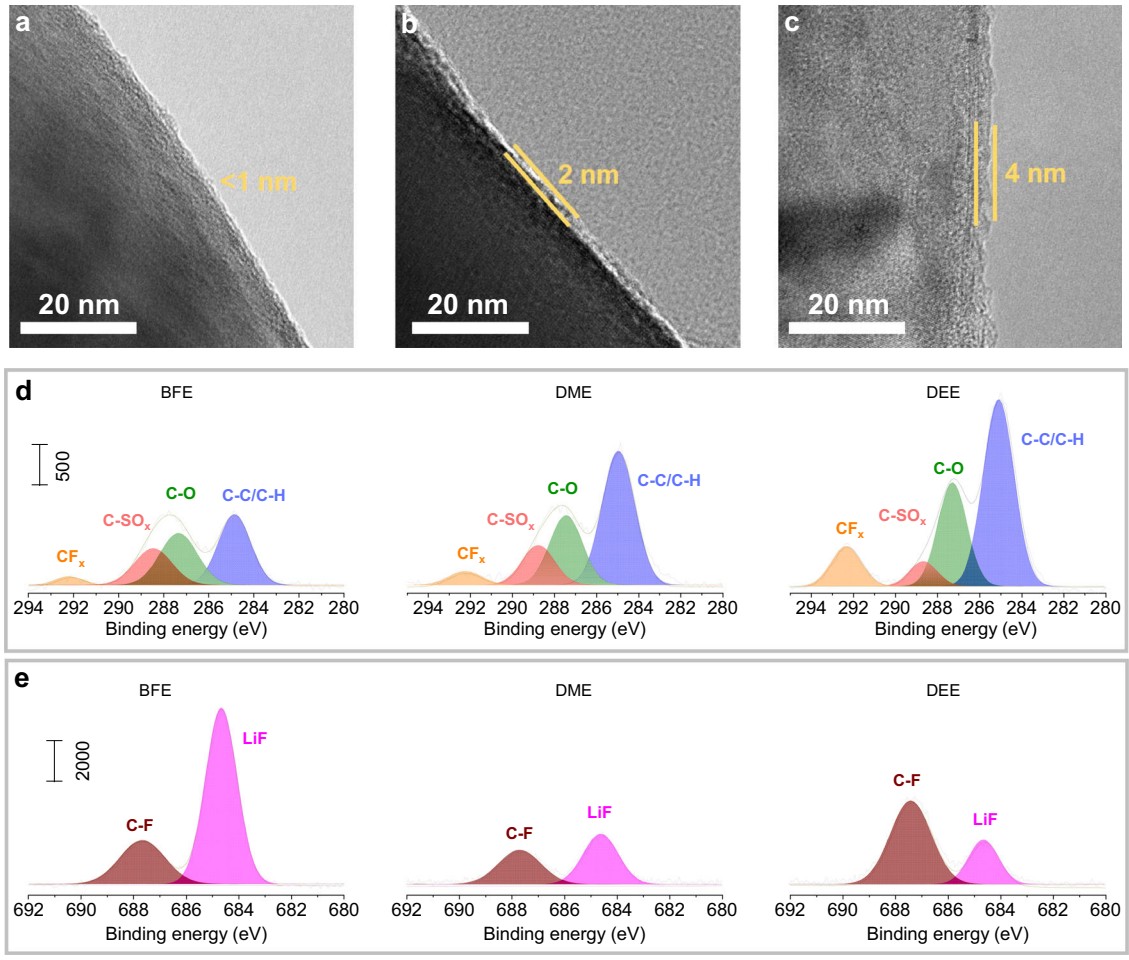

**Fig. 6 | Ex situ interface characterizations of cycled NCM811-based electrodes.** **a-c**, High-resolution TEM images of the cycled NCM811 cathode in BFE (**a**), DME (**b**), and DEE (**c**) electrolytes. **d**, **e**, C1s (**d**), and F1s (**e**) XPS spectra of the cycled NCM811 cathode in different electrolytes (BFE, DME, and DEE, coin cells after 50 cycles with a fully discharged (2.8 V) state at 30 °C).

## Low-temperature electrochemical characterizations of Li metal cells with monofluoride ether-based electrolyte

To understand the effect of the temperature on the Li metal cycling for cells with monofluoride ether-based electrolytes, we carried out several electrochemical characterizations. The average CE of Li metal with BFE and DEE electrolytes is estimated by Aurbach's Li∥Cu cells at a low temperature of −30 °C. The BFE electrolyte offers a Li metal CE of up to 99.5% with a low overpotential of 15 mV during the plating/stripping process, while the Li metal CE in the DEE electrolyte is 98.8% with a relatively large overpotential of 120 mV (Fig. 7a, Supplementary Fig. 36). The improved CE of Li metal in BFE electrolyte should be ascribed to uniform and dense deposition morphologies of Li metal under low-temperature conditions (Fig. 7b). In comparison, various Li metal porous regions are observed and some dendritic deposits are also formed at the low temperature of −30 °C. Moreover, Li∥NCM811 coin cells (areal capacity: 3.5 mAh cm⁻², N/P: 2.8, and electrolyte usage: 2.4 g Ah⁻¹) are assembled and tested with BFE, DME, and DEE electrolytes at various temperatures ranging from −60 to 60 °C. As presented in Fig. 7c, the coin cell of Li∥NCM811 with our BFE electrolyte delivers capacities of 90, 139, 160, 181, 202, and 226 mAh g⁻¹ at −60, −30, −20, 0, 30, and 60 °C, respectively. The Li∥NCM811 coin cell can be charged and discharged at a low temperature of −60 °C and maintains >45% of the capacity at 30 °C (current density: 0.35 mA cm⁻²). At the same temperature of −60 °C, the full cell with DEE electrolyte barely retains 8% of the capacity at 30 °C. In addition, the cell with

DEE electrolyte suffers a rapid capacity decay and fluctuation at 60 °C due to its low boiling point (at 1.75 mA cm⁻²). For the DME electrolyte, the Li∥NCM811 coin cell shows slightly lower reversible capacities than that of the BFE electrolyte at 30 and 60 °C but fails to operate at relatively low temperatures below −20 °C due to the complete solidification of the electrolyte (Supplementary Fig. 37). Even at −30 °C, the Li∥NCM811 coin cell also exhibits cycling stability with capacities retention of >90% during 150 cycles (at 1.75 mA cm⁻², Fig. 7d). Despite the good low-temperature performance of DEE electrolyte[16,32], the Li∥NCM811 coin cell suffers a severe capacity degradation and fails after 50 cycles.

To highlight the performance of our Li∥NCM811 pouch cells with BFE electrolyte, the specific energy is analyzed at various current densities and temperatures. Based on the total weight of the entire cell (including the current collectors, electrodes, separators, and electrolytes), current state-of-the-art LMBs report specific energies >300 Wh kg⁻¹ at current densities <1 mA cm⁻² (Fig. 7e)[20–26,32–37]. However, the high specific energy of most Li metal cells s is only available at 30 °C and reduced (<100 Wh kg⁻¹) when the current density is increased to 10 mA cm⁻². At low-temperature or high-rate conditions, the specific energy of LMBs is reduced[16,23,32–39]. Compared to previously reported Li metal and Li-ion batteries, the Li∥NCM811 punch cell with BFE electrolyte could provide higher specific energy (267 ~ 426 Wh kg⁻¹) at a wide range of current densities (0.7 ~ 17.5 mA cm⁻²) and temperatures (−60 ~ 60 °C) (Fig. 7e, f).

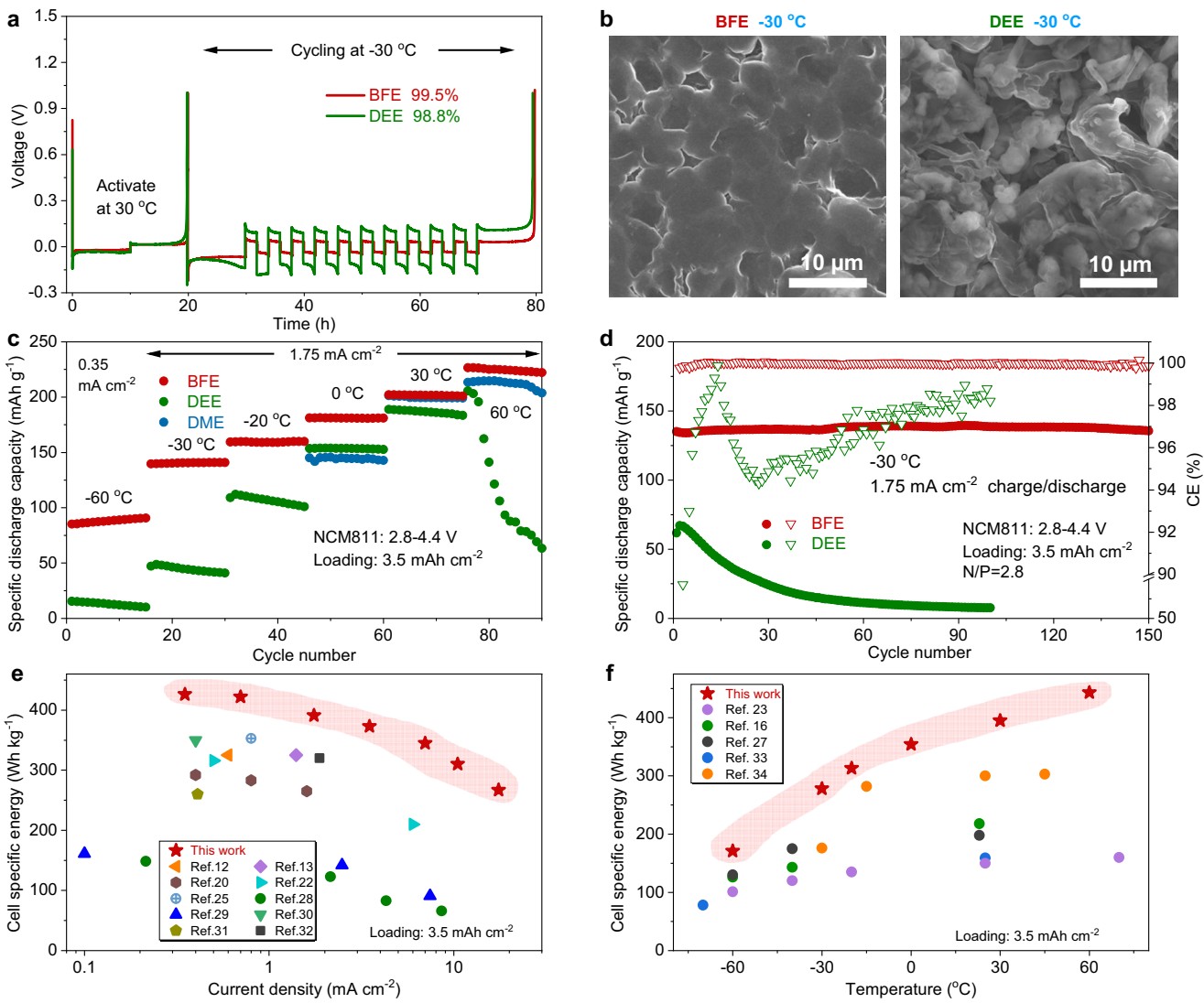

**Fig. 7 | Low-temperature electrochemical energy storage characterizations of the monofluoride ether-based electrolyte in various Li metal cell configurations. a** Li plating/stripping CEs evaluated via Aurbach's measurement at −30 °C using Li||Cu coin cells. **b** SEM images of plated Li on Cu foil in BFE and DEE electrolytes at −30 °C, respectively. **c** Capacities of Li||NCM811 with BFE, DEE, and DME electrolytes at different temperatures varied from −60 to 60 °C. **d** Long cycling performance of Li||NCM811 coin cells using BFE and DEE electrolytes at a low temperature of −30 °C. **e, f** Specific energies of LMBs at various current densities (**e**) and different temperatures (**f**).

In summary, we designed and synthesized a monofluoride ether solvent to prepare non-aqueous electrolyte solutions with fast Li⁺ ions transport with high oxidation stability for fast-charging and low-temperature lithium metal batteries under practical conditions. We demonstrated that the monofluoro substitution could increase the ion conductivity of fluorinated electrolytes by maximizing the coordination interaction of fluorine atoms with Li⁺ cations. Due to the strong Li-F and Li-O tridentate coordination interactions, the single-salt and single-solvent monofluoride electrolytes show high ionic conductivity, improved Li metal cyclability, and oxidation stability within wide temperature ranges. Electrochemical tests of 50-µm-thick Li||high-loading NCM811 coin cells with monofluoride electrolytes exhibit an areal capacity of 3.5 mAh cm⁻², good cycling performance at 17.5 mA cm⁻² and stable low-temperature (−30 °C) operation over 150 cycles. Moreover, we also assembled and tested a practical 320-mAh-level Li||NCM811 pouch cell (with a technology readiness level of 4)[31], which delivered an initial specific energy of 426 Wh kg⁻¹ (based on the weight of the entire cell) and capacity retention of 80% after 200 cycles at charge/discharge rates of 0.8/ 8 mA cm⁻².

## Methods
### Materials
2-Fluoro-ethanol, 2,2-difluoro-ethanol, 2,2,2-trifluoro-ethanol, 1-bromo- 2- fluoro-ethane, p-toluene sulfochloride, bis(2,2,2-tri-fluoroethyl) ether (BTFE), and potassium hydroxide (KOH, 95%) were purchased from Sigma-Aldrich. Other general reagents, including N-methylpyrrolidone, diethyl ether (DEE), anhydrous magnesium sulfate (MgSO₄), calcium hydride (CaH₂), and lithium hydride (LiH), were purchased from Aladdin Bio-Chem Technology Co. Ltd (Shanghai, China). All chemicals were used without further purification. Lithium bis(fluorosulfonyl)imide (LiFSI), 1,2-dimethoxyethane (DME), LiNi₀.₈Co₀.₁Mn₀.₁O₂ (NCM811, 3.5 and 4.0 mAh cm⁻²) cathode sheets were provided by Shenzhen CAPCHEM Technology Co. Ltd. The commercial Li-battery separator Celgard 2400 (20 µm thick) was purchased from Celgard and used in all coin and pouch cells. Thick Li foil (400 µm) and thin Li foil attached on 10 µm copper meshes (40 and 50 µm) were purchased from China Energy Lithium Co. Ltd (Tianjin, China). The thin Cu current collectors (12 µm) and other battery materials were all purchased from Shenzhen Kejing Star Technology Co. Ltd.

## Synthesis

Bis(2-fluoroethyl) ether (BFE, Fig. 1c). To a round-bottom flask, 70.8 g KOH (1.2 mol) was added, and 64 g 2-fluoro-ethanol (1 mol) in 60 mL N-methyl pyrrolidone (NMP) was then added to the above flask dropwise. After addition, the mixture was cooled below −5 °C followed by adding 127 g 1-bromo-2-fluoro-ethane (1 mol) dropwise under stirring. The mixture was then stirred at −5 °C for 2 h and then placed at 30 °C for 24 h. After the reaction, the precipitated white solid was removed through filtration, and the residual solution was extracted with ethyl ether, followed by washing with brine three times. The obtained solution was dried through anhydrous $MgSO_4$ and $CaH_2$ and concentrated via rotary evaporation to give a crude product. The final high-purity BFE ether was collected by atmospheric distillation around 128-135 °C and then stored in a brown glass bottle containing a 4 Å molecular sieve. The water content was detected to be <10 ppm by the Karl Fischer method. Yield: 82%. $^1H$ NMR and $^{19}F$ NMR are shown in Fig. 1d.

2,2-difluoroethyl-2-fluoroethyl ether (DFE, Supplementary Fig. 11a): The synthetic protocol is similar to that of BFE, except that the 2-fluoro-ethanol was replaced with 2,2-difluoro-ethanol, and the final distillation temperature is controlled at 120-130 °C. The water content was detected to be <10 ppm by the Karl Fischer method. Yield: 80%. $^1H$ NMR and $^{19}F$ NMR were shown in Supplementary Fig. 12a, b, respectively.

Bis(difluoroethyl) ether (BDE, Supplementary Fig. 11b): 2, 2-difluoroethanol (41 g, 0.5 mol) was dissolved in THF (130 mL), and the solution was cooled down to 0 °C. A 6 M aqueous NaOH (130 mL) solution was added dropwise over 15 min, followed by the dropwise addition of a p-tosyl chloride (102 g, 0.54 mol) solution in THF (140 mL) over 30 min. The reaction mixture was further stirred at 0 °C for 1 h, warmed to 25 °C and then stirred for another 2 h. The resulting solution was diluted with ethyl ether (500 mL) and 1 M NaOH aqueous solution (400 mL). After separating the aqueous layer, the organic phase layer was washed three times with brine (100 mL), dried by anhydrous $MgSO_4$, and concentrated via rotary evaporation to produce 2, 2-difluoroethyl tosylate as a colorless liquid. Yield: 114.0 g (0.485 mol, 97%).

2, 2-difluoroethyl tosylate (114.0 g, 0.485 mol) and 2, 2-difluoroethanol (27 g, 0.32 mol) were dissolved in 100 mL n-methyl pyrrolidone (NMP), the solution was cooled down to 0 °C, then 45 wt% KOH aqueous solution (95 mL) was added dropwise under stirring. Afterward, the solution was warmed to 50 °C and stirred for 5 h. A large amount of white solid precipitated from the solution and the mixture was heated to 70 °C for 2 h to remove the residual tosylate. After cooling the reaction to 25 °C, the white solid was removed via filtration, and the residual solution was extracted with ethyl ether, followed by washing it three times with brine. The obtained solution was dried through anhydrous $MgSO_4$, $CaH_2$, and LiH, respectively. Then, the dried solution was concentrated via rotary evaporation to give a crude product. The final high-purity bis(2,2-difluoroethyl) ether was collected by atmospheric distillation around 115-125 °C. Yield: 84%. $^1H$ NMR and $^{19}F$ NMR were shown in Supplementary Fig. 12c, d, respectively.

2,2,2-trifluoroethyl 2-fluoroethyl ether (TFEE): The synthetic protocol is similar to that of BFE, except that the 2-fluoro-ethanol was replaced by 2,2,2-trifluoro-ethanol (Supplementary Fig. 11c), and the final distillation temperature is controlled at 100-105 °C. The water content was detected to be <10 ppm by the Karl Fischer method. Yield: 76%. $^1H$ NMR and $^{19}F$ NMR were shown in Supplementary Fig. 12e, f, respectively.

## Electrolytes

Commercial lithium salts of LiFSI (1,870 and 3,740 mg) were dissolved into 10 mL solvents (e.g., DEE, DME, and BFE) to directly obtain 1 M and 2 M electrolytes, respectively. It is worth noting that the saturated salt concentration is roughly 0.2 M in TFEE electrolyte due to the insufficient solubility to LiFSI. All the electrolytes were prepared and stored in an argon-filled glovebox (oxygen <0.01 ppm, water <0.01 ppm).

## Physicochemical characterizations

The chemical structures of the solvents and corresponding electrolytes were analyzed via nuclear magnetic resonance (NMR) (400 MHz, Bruker). The densities of solvents were tested using a DensitoPro densimeter (Mettler Toledo) at 30 °C. For the NMR test of solvents, $CDCl_3$ was used as a solvent to lock fields. The trifluoroacetic acid ($CF_3COOH$) has been selected as the reference sample for the calibration of fluorinated electrolytes by using coaxial nuclear magnetic tubes. All the NMR measurements were performed with a decoupling mode. The morphology of plated metallic Li was characterized using a field emission scanning electron microscope (SEM, TASCAM MIRA3). To avoid contact with air, the plated Li samples on Cu foils were prepared in the glove box and transferred using a seal transfer bin with the protection of Ar. The ionic conductivities of electrolytes were measured via a FiveGo F3 conductivity meter (Mettler Toledo) at different temperatures. Electrolyte viscosities were measured using an Ares G2 rheometer (TA Instruments) with an advanced Peltier system at 25 °C. X-ray photoelectron spectroscopy (XPS) was obtained through an AXIS-ULTRA DLD spectrometer (Shimazu-Kratos). The XPS spectra are calibrated by using C1s (284.8 eV) as the reference peak. The etching depth for the sputtering is 6 nm/min. The characterization of Li metal samples was performed on an aberration-corrected FEI Krios G3i microscope with Gatan Continuum (1069) EELS spectrometer and Falcon 3 direct detection device. The automatic liquid nitrogen perfusion system can automatically maintain the low temperature of the sample chamber and the lens barrel for several days to ensure high imaging stability. The resolution limit of the microscope can reach -0.14 nm. The samples in different electrolytes were prepared by electrochemically plating Li metal on the copper grid (400 mesh) at $0.25\,mA\,cm^{-2}$ for 2 h using coin cells. The extracted Li metal samples were immediately frozen in an improved argon-filled glove box and then transferred to the Cryo-TEM chamber. The cryo-TEM images were acquired with an electron dosage of $120\,e\,Å^{-2}$ and the Cryo-EELS are acquired using 11 pA current with 0.1 s dwell time at each pixel for core edges.

## Theoretical calculations

The molecular geometries for the ground states were optimized by density functional theory (DFT) at the B3LYP/6-31 G (d, p) level, and then the energy, orbital levels, and electrostatic potentials (ESPs) of the molecules were evaluated. All the DFT calculations were carried out with the Gaussian 09 package. Binding energies of the $Li^+(Solvent)_x$ complexes were calculated after geometry optimizations, in which the full complexes were optimized with and without $Li^+$, representing their separation at an infinite distance. The binding energy was calculated as: $E_d = E_{Li^+(solvent)x} - (E_{Li^+} + E_{x(solvent)})$[40,41].

All-atom molecular dynamics (MD) simulations: MD simulations were performed using the GROMACS 5.1.4 package[42]. The optimized potentials for a liquid simulations all-atom (OPLS-AA) force field were adopted to describe the interatomic interactions of the electrolyte system. Partial atomic charges were optimized by DFT. According to the molar ratio in experiments, 1000 LiFSI molecules and 4500 solvent molecules (2 M DEE and 2 M BFE) were randomly put into the cubic simulation box by using Packmol software[43]. The steepest descent method was applied to minimize the energy of the system. Then, 10 cycles of quench-annealing dynamics between 298 and 698 K were conducted to eliminate the persistence of meta-stable states. After that, a 50 ns MD simulation in the isothermal-isobaric ensemble was conducted at the temperature of 303 or 243 K and the pressure of 1 bar, and the last 20 ns MD trajectory was used for analysis. During simulations, the temperature was controlled by the Nosé-Hoover

thermostat algorithm with a coupling constant of 0.2 ps, and the pressure was controlled by the Parrinello-Rahman algorithm. The LINCS algorithm was employed for bond constraints. The long-range electrostatic interactions were treated with the Particle Mesh Ewald (PME) method. The non-bonded potential truncation was performed with the cut-off radius of 12 Å for the Lennard-Jones potential. Periodic boundary conditions were used in all three directions. The time step was set at 1 fs. The visualization and analysis of simulation results were performed using VMD and internal codes[44].

### Electrochemical measurements

All battery components used in this work were commercially available and all electrochemical tests (except where indicated) were carried out using 2032-type coin cells. All cells were fabricated in an argon-filled glovebox ($H_2O$ < 0.1 ppm, $O_2$ < 0.1 ppm), and one layer of Celgard 2400 was used as a separator for all batteries. The tests of $Li^+$ transference number (LTN) and linear sweep voltammetry (LSV) were carried out on a Solartron electrochemical workstation (1260 A). The cycling tests for coin cells were carried out on Neware system. For the LTN measurements, a 10-mV constant voltage bias was applied to the Li||Li cells. Linear LSV tests were conducted over a voltage range from open circuit to 5 V. For Li||Cu cell CE cycling tests, five pre-cycles between 0 and 1 V were initialized to minimize the side reaction between the Li and Cu electrode surface, and then cycling was done by depositing Li onto the Cu electrode; the Li was then stripped to 1 V at different current densities. The average CE was calculated by dividing the total stripping capacity by the total deposition capacity after the formation cycle. For the Aurbach's CE test, a standard protocol was followed: (1) performed one initial formation cycle with Li metal deposition of 5 mAh cm$^{-2}$ on Cu with 0.5 mA cm$^{-2}$ and stripping Li to 1 V; (2) deposited 5 mAh cm$^{-2}$ Li on Cu with 0.5 mA cm$^{-2}$ as a Li reservoir; (3) repeatedly stripped/deposited Li of 1 mAh cm$^{-2}$ with 0.5 mA cm$^{-2}$ for 10 cycles; (4) stripped all Li to 1 V. The Li|| NCM811 cells (cathode capacity: 3.5 mAh cm$^{-2}$, Li foil: 50 μm) with different electrolytes were cycled between 2.8 and 4.4 V at different rates after the first two activation cycles at 0.35 mA cm$^{-2}$. The NCM811 cathode includes 97 wt% active materials, 1 wt% Super P, and 2 wt% PVDF binders. For the Li||NCM811 pouch cells, two cathode sheets (double side, 2.7*7.4 cm$^2$) with a high areal capacity of 4.0 mAh cm$^{-2}$ were selected to pair with three 40-μm-thick Li foils deposited on Cu meshes. The anode-free cell was assembled by paring NCM811 cathode with Cu foil and tested with the current density of 0.35 mA cm$^{-2}$ at 30 °C. The graphite (Gr) anode with an areal capacity loading of 3.7 mAh cm$^{-2}$ includes 95 wt% active materials, 2 wt% Super P, and 3 wt% binders. The Li||Gr coin cells were cycled between 0.01 and 2 V at 1.85 mA cm$^{-2}$ and 30 °C. The electrochemical impedance spectroscopy (EIS) of Li||NCM811cells was performed from 10 MHz to 10 mHz at an amplitude of 10 mV using a Solartron 1287 electrochemical workstation. Before carrying out the EIS measurements at 30 °C, cells were discharged to 3.9 V. For all electrochemical measurements, three cells were measured at the same condition. The specific current and specific capacity refer to the mass of the active material in the positive electrode. All cells were prepared in an argon-filled glovebox ($H_2O$ < 0.1 ppm, $O_2$ < 0.1 ppm) and tested in constant-temperature chambers.

### Reporting summary

Further information on research design is available in the Nature Portfolio Reporting Summary linked to this article.

## Data availability

The data that supports the findings of this study are available from the corresponding author upon request.

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

## Acknowledgements

We acknowledge the support from the Key-Area Research and Development Program of Guangdong Province (2020B090919001), National Natural Science Foundation of China (22109066 and 22078144), Natural Science Foundation of Guangdong Province (2022A1515011005 and 2023A1515010686), Guangdong Basic and Applied Basic Research Foundation (2019A1515110881 and 2020A1515110300), Shenzhen Science and Technology Program (JCYJ20220818100218040), and the Science and Technology Planning Project of Guangdong Province (2021A0505110001).

## Author contributions

G.Z., J.C., J.L., T.W., and Y.D. conceived the concept and designed the experiments. G.Z. and J.C. contributed to this work in the experimental planning, experimental measurements, data analysis, and manuscript preparation. J.-W.L. conducted the MD simulations and DFT calculations. R.W., G.S., and K.Y. participated in material synthesis and characterization. L.W., W.H., C.W., and H.Z. assisted in the manuscript preparation. J.C., J.L., T.W., and Y.D. co-wrote the manuscript. All authors discussed the experimental results and commented on the manuscript.

## Competing interests

The authors declare no competing interests.
