## [Peer Review File · Nature Communications]

REVIEWER COMMENTS

Reviewer #1 (Remarks to the Author):

This manuscript elucidates fast-charging and low-temperature LMBs through ether solvent fluorination, proposing monofluoride bis(2-fluoroethyl) ethers (BFE) with enhanced ionic conductivity as well as oxidation stability. The authors adopted other ether-based solvents with various number of fluorine functional groups (i.e. BTFE, BDE, TFFE, DFE, and DEE with no fluorine group) and compared the ionic conductivity, solvating environment, and following cell performance.

Through NMR analysis with molecular calculation and simulation, the authors identified that monofluoro substituent (-CH₂F) could induce the strong (Li-F and Li-O) tridentate coordination interaction with Li⁺, leading high solvation capability and maximized ion conductivity. This gives a clue for solving the problem for multi-fluoro group which largely reducing the electron cloud density of ion-conducting groups, leading weaken salt dissociation ability and following sluggish ion transport. With matching up BFE and other fluorine-substituted solvents, plausible strategy for relationship between fluorination degree and ionic conductivity as well as solvating ability is well clarified. With retaining oxidation stability from electron-withdrawing fluorine group, the high battery performance is also sufficiently proposed in industrial scale, NCM811 high loading, and low-temperature conditions. Overall, I would be supportive of this work. The detailed comments are as below:

1. It is necessary to explain the crucial point of ionic conductivity and solvation structure for the lithium metal interface. Forming a stable lithium metal interface in a lithium metal battery is an important part. According to the author, BFE has relatively high ionic conductivity (~7 mS cm⁻¹) compared to various electrolytes (DEE, DFE, TFFE), and it is claimed that it brings a stable electrodeposition shape at the lithium metal interface. Although 1M LiFSI DME has an ionic conductivity of ~16.9 mS cm⁻¹ (Nature Communications 6, 6362, 2015), dendrites are severely generated on the lithium surface as the cycle increases. It is questionable whether simply high ionic conductivity can bring stability to the lithium metal interface.

In addition, if the BFE solvent performs strong solvation on Li⁺ ions, there are likely to be many SSIP structures compared to other comparison electrolytes. Analysis of the solvation structure through Raman spectroscopy is required. If there are many SSIP structures, it is necessary to explain why the lithium interface is stabilized.

2. Interfacial analysis data of SEI layer of cycled lithium metal is required. For rapid charging, it is important not only to ionic conductivity but also how uniformly the interface is well formed. XPS data is

needed to confirm what kind of interfacial components are formed when using an electrolyte with BFE compared to DEE, and if possible, it would be good to check the uniformity through the depth profile.

Reviewer #2 (Remarks to the Author):

The manuscript titled "A monofluoride electrolyte with high ionic conductivity for fast-charging and low-temperature lithium metal batteries" presented a new fluorinated electrolyte combining light weight, low viscosity, oxidative stability, and high ionic conductivity. Owing to the monofluoride molecular design, the BFE electrolyte can achieve the high ionic conductivity under wide-temperature conditions, Li metal cycling efficiency (99.75 %), and oxidation stability (4.7 V vs. Li/Li+.) The BFE-based electrolyte demonstrated excellent cycling performance of pouch cell (areal capacity: 4 mAh cm⁻², N/P: 2, and electrolyte: 2.4 g Ah⁻¹) with energy density (426 Wh kg⁻¹) and high-capacity retention (>80%) during 200 cycles. The full cell with BFE electrolyte also exhibits stable low-temperature (-30 °C) operation over 150 cycles. However, systematic characterization for the newly proposed high-voltage ether-based solvent is still needed, such as Al corrosion, XPS depth profiling studies of SEI and CEI layers, flammability, long-cycling symmetric Li-Li cell performance and EIS, which is very important for the interface study and realistic Li metal full cell.

Overall, I would suggest reconsider after major revision. All the comments are shown as below.

1. The XPS depth profiling studies can reveal the SEI and CEI composition and distribution along the thickness. Can the authors run XPS analysis of the cycled Li-NCM811 full cells with different electrolytes to study the SEI and CEI?
2. Flammability is one of the critical safety issues for LMBs. After decreasing the fluorine amount of the solvent, how about the flammability of the BFE solvent? Please compare the flammability of all fluorinated molecules.
3. SEI on Li anode should be characterized by Cryo-EM and also FFT to reveal the SEI composition. The CEI in full cells after cycling also need be analyzed by HR-TEM to check the phase transition or surface lattice reconstruction on NCM811 particles.
4. For the average CE by Aurbach method, can the authors provide reproduced data (three more) for the BFE (99.75%)? I do not expect the reproduced cells to be as high as 99.75% but some values >99.65% are definitely needed to solidify this outstanding performance.
5. CC-CV charge method is widely used in the real battery. Can the authors run the Li-NCM811 full cells through a constant-current-constant-voltage protocol at 4.4V (Nature Energy 7, 94–106 (2022)) at 0.3 C to check the long-cycling performance using BFE electrolyte?

6. Li|Li symmetric cells are widely used to evaluate interfacial stability. Long-cycling of Li|Li symmetric cells is required for realistic batteries. Can the authors run the Li|Li symmetric cells at 1 mA cm⁻², 1 mAh cm⁻² for long-cycling until cell is failed?
7. LSV in Li|Al cell is not enough convinced in real battery due to the Al surface passivation and low surface area, can the authors provide LSV results using Super-P-PVDF and Pt electrodes? And the Li|Al cell holds for several hours with increasing voltage (Nature Energy 5, 526–533 (2020))?
8. Al corrosion caused by FSI anion is serious in the real battery. Can the authors using Li|Al half cell hold at 5 V for 48 hours with different electrolyte and check the Al foil by SEM and XPS?
9. Did the authors try the 3M LiFSI-BFE? Were the various performance better than the 2M LiFSI-BFE?
10. EIS is essential for the interface study in the LMBs, can the authors conduct the systematic EIS measurement and analysis for the different electrolytes?
11. Author mentioned BFE can coordinate Li⁺ ions with one Li-O and two Li-F interaction. Can author grow single crystal and check and diffraction to prove the unique coordination with BFE? Follow the reference Nature Energy 5, 526–533 (2020).
12. BFE solvents exhibited the high CE of 99.75%. Can author measure the performance of anode-free full cell to compare with other literatures?

Reviewer #3 (Remarks to the Author):

This work reports the design, synthesis, and application of a novel monofluoride ether, i.e., bis(2-fluoroethyl) ethers, as an electrolyte solvent for high-energy and high-power lithium metal batteries with wide operative temperature ranges. The electrochemical performance is very impressive, particularly at low temperatures. Some experiments and simulations regarding the coordination structure and Li⁺ transport have been conducted to interpret the remarkable performance. The results and findings in this work are certainly interesting and important. However, a series of concerns do emerge when the reviewer goes through the article. Moreover, the reviewer has a feeling that the work is not well finished, as some important aspects relevant to the good performance have not been investigated. Therefore, an appropriate decision can only be given after the re-valuation of this article upon a major revision. My detailed comments are included below:

1. The first concern is the cost. It is claimed that the bis(2-fluoroethyl) ethers (BFE) is synthesized with only two basic, low-cost reagents, i.e., 2-fluoroethanol (CAS: 371-62-0) and 1-bromo-2-fluoroethane (CAS: 762-49-2). But this is not true. The prices of 2-fluoroethanol and 1-bromo-2-fluoroethane are 166 euro/ 25 g and 160 euro/ 25 g, respectively (prices from VWR). The authors mentioned that these chemicals were purchased from Aldrich Sigma, but the reviewer did not find them from the web

shop. Although the electrolyte leads to the very good performance even under practical conditions, the high cost of the solvent clearly hinders the applications.

2. Why the oxygen from DEE shows lower electron density than the oxygen from DFE and BFE (Figure 2a)? Due to the electron withdrawing effect, the oxygen from DFE and BFE should exhibit lower electron density than that from DEE. Should not be like this?

3. The authors provide enough results proving the coordination between Li and F from CH₂F and demonstrating the effect of fluoride content on the Li⁺-F coordination. But the experimental characterization on the coordination of Li⁺ to FSI⁻ and to oxygen from ether is weak or even not mentioned. In the latter MD simulation parts, the authors mentioned that the aforementioned, not well-characterized coordination also has effects on the Li⁺ transport. Therefore, this reviewer recommends the authors to conduct Raman spectra to prove the statements in the MD simulation, and to show the influence of the different fluorinated group on the coordination of Li⁺ to FSI⁻ and ether oxygen.

4. How the samples were prepared for NMR measurements, how the chemical shift was calibrated, and what reference was used for the calibration are not mentioned in the experimental section. This leads to the concern on the reliability of some of the results, e.g., some ¹⁹F NMR spectra (Fig. 2b, and S5), ⁷Li NMR spectra (Fig. 2e) and ¹⁷O NMR spectra (Fig. S6), as only single peak appears in each spectrum.

5. Fig. 2f shows the solvating energy. How about the contribution from Li⁺-F and Li⁺-O? From the MD simulation (Fig. 3e), the coordination of Li-OBFE is clearly stronger or more dominant than Li-FBFE, while the characterization mainly focuses on the coordination between Li⁺ and F from the solvents. What is the respective role of Li⁺-O_{solvent} and Li⁺-F_{solvent} in the proposed solution structure and enhanced Li⁺ transport?

6. The Li⁺ transference number was measured but lead to some concerns. In Fig. S2, the impedance before and after the polarization is very different, and the current does not finally reach a steady state, which affect the reliability of the obtained results. In Fig. S3, the impedance of the cell employing DEE is five times of that with BFE. But the current of DEE cell is still much higher than that of BFE cell, specifically, 6 times. This is clearly wrong. In fact, the impedance from the interphase is more than 100 times of that from the electrolyte at -30 °C, the adopted electrochemical method is not suitable to get reliable Li⁺ transference number.

7. The rate capability and low temperature performance not only relates to the Li⁺ transport in the electrolyte but also the interphases. However, the characterization of the interphase is not provided. Therefore, XPS characterizations on the SEI on Li metal should be provided to demonstrate the influence of the fluorinated groups.

8. Apart from the very good rate capability, the author reported super high columbic efficiency up to 99.75% at room-temperature and 99.5% at -30 °C. These values refresh the records under the same protocol. Three duplicated cells are recommended to demonstrate the reproducibility.

9. Compared to DEE and DME-based electrolytes, BFE-based electrolyte brings to very good cyclability of Li||NMC811 cells, which clearly relates to a more stable cathode/electrolyte interphase on NMC811. Nonetheless, this is not characterized in the manuscript. To make the work more completed, XPS characterizations on the cathode-electrolyte interphase should be conducted.

Point-to-point Reply to reviewers' Comments

Reviewer 1:

This manuscript elucidates fast-charging and low-temperature LMBs through ether solvent fluorination, proposing monofluoride bis(2-fluoroethyl) ethers (BFE) with enhanced ionic conductivity as well as oxidation stability. The authors adopted other ether-based solvents with various number of fluorine functional groups (i.e. BTFE, BDE, TFFE, DFE, and DEE with no fluorine group) and compared the ionic conductivity, solvating environment, and following cell performance. Through NMR analysis with molecular calculation and simulation, the authors identified that monofluoro substituent (-CH₂F) could induce the strong (Li-F and Li-O) tridentate coordination interaction with Li⁺, leading high solvation capability and maximized ion conductivity. This gives a clue for solving the problem for multi-fluoro group which largely reducing the electron cloud density of ion-conducting groups, leading weaken salt dissociation ability and following sluggish ion transport. With matching up BFE and other fluorine-substituted solvents, plausible strategy for relationship between fluorination degree and ionic conductivity as well as solvating ability is well clarified. With retaining oxidation stability from electron-withdrawing fluorine group, the high battery performance is also sufficiently proposed in industrial scale, NCM811 high loading, and low-temperature conditions. Overall, I would be supportive of this work. The detailed comments are as below:

Response: We appreciate the reviewer's very positive and constructive comments for our manuscript. The followings are the details of the responses. According to the reviewer's suggestion, the solvation structure of BFE electrolyte and the interfacial SEI components of cycled Li metal is systematically analyzed through Raman spectroscopy, and Cryo-transmission electron microscopy (Cryo-TEM), and X-ray photoelectron spectroscopy (XPS), respectively. The related data and discussion part are added to the main text (marked by red color). We believe that all the requirements for improving our manuscript have been satisfied.

1. It is necessary to explain the crucial point of ionic conductivity and solvation structure for the lithium metal interface. Forming a stable lithium metal interface in a lithium metal battery is an important part. According to the author, BFE has relatively high ionic conductivity ($\sim 7 \text{ mS cm}^{-1}$) compared to various electrolytes (DEE, DFE, TFFE), and it is claimed that it brings a stable electrodeposition shape at the lithium metal interface. Although 1M LiFSI DME has an ionic conductivity of $\sim 16.9 \text{ mS cm}^{-1}$ (Nature Communications 6, 6362, 2015), dendrites are severely generated on the lithium surface as the cycle increases. It is questionable whether simply high ionic conductivity can bring stability to the lithium metal interface. In addition, if the BFE

solvent performs strong solvation on Li^+ ions, there are likely to be many SSIP structures compared to other comparison electrolytes. Analysis of the solvation structure through Raman spectroscopy is required. If there are many SSIP structures, it is necessary to explain why the lithium interface is stabilized.

Response: According to the reviewer's suggestion, we performed the characterization of the solvation structures of various electrolytes (BFE, DEE, and DME) through Raman spectroscopy. It has been recognized that the DME and DEE electrolytes show typical solvation structures of solvent-separated ion pairs (SSIPs) and contact ion pairs (CIPs), respectively. This result is accorded with the previous report (Nature Energy 2021, 6, 303-313). The S-N-S bending peak of the FSI^- anions in the DEE electrolyte undergoes a small shift from 774 to 750 cm^{-1} after the salt dissolution, suggesting the strong Li^+/FSI^- the interaction of CIP structures. In comparison, the S-N-S bending peak of the FSI^- anions is markedly shifted to 722 cm^{-1} after dissolving the salts into the DME solvents, indicating the weak Li^+/FSI^- the interaction of SSIP structures. This result is also consistent with our MD simulations. The BFE electrolyte exhibits a moderate bending peak (735 cm^{-1}) of the FSI^- anions between DME and DEE electrolytes, indicative of the existence of SSIP and CIP solvation structures. The SSIP structure of the BFE electrolyte could substantially improve the overall ion conductivity by promoting the effective dissociation of the LiFSI salt. On the other hand, the CIP structure of BFE electrolytes could stabilize the interface of Li metal anode and high-voltage cathode by decomposing the anions in solvation structures and forming LiF components. The discussion part has been added in the main text (marked by red color). These results are also confirmed by the molecular dynamics simulations in Fig. 3. The Raman spectrum data is added as Supplementary Fig. 17.

Note, the data is added as Supplementary Fig. 17.

2. Interfacial analysis data of SEI layer of cycled lithium metal is required. For rapid charging, it is important not only to ionic conductivity but also how uniformly the

interface is well formed. XPS data is needed to confirm what kind of interfacial components are formed when using an electrolyte with BFE compared to DEE, and if possible, it would be good to check the uniformity through the depth profile.

Response: According to the reviewer's suggestion, the deposit morphology and SEI composition of Li metal anode after cycling were well studied using scanning electron microscopy (SEM), X-ray photoelectron spectroscopy (XPS), and Cryo transmission electron microscopy (Cryo-TEM). Here, 5 mAh cm⁻² of Li metal is electrodeposited onto bare Cu foil to examine the deposition morphology and solid electrolyte interphase (SEI) components of Li metals in various electrolytes, including BFE, DME, and DEE. The scanning electron microscopy (SEM) image reveals smooth and compact deposit morphologies of Li metal in BFE electrolyte (Fig. 5a). The compact packing and large crystal size of Li deposits are beneficial to the improvement of cycling stability by reducing the formation of thick SEI and dead Li. However, large amounts of porous Li deposits in both DME and DEE electrolytes easily induce continuous SEI and dendritic growth (Fig. 5b and 5c). At the same plating capacity, the plated Li metal layer in the BFE electrolyte is much thinner than DME and DEE electrolytes (Fig. 5d-5f). The SEI composition of Li metal anode after repeated plating/stripping cycles in BFE electrolyte was also analyzed by low-dose Cryo transmission electron microscopy (Cryo-TEM) (Fig. 5g-5i). A dense and uniform SEI layer with a small thickness of 20 nm is observed in Cryo-TEM images. The inverse fast Fourier transform (IFFT) characterization displays the magnified lattice spacing of 0.20 nm is well-matched with the (200) crystal plane of LiF (Fig. 5h) and the lattice spacing of 0.27 nm is accorded with the (111) crystal planes of Li₂O (Fig. 5i). This result is also confirmed by the corresponding electron energy loss spectroscopy (EELS) spectra (Supplementary Fig. 30). In addition, X-ray photoelectron spectroscopy (XPS) with etching depth profiles were also used to analyze the SEI components of Li metal in various electrolytes (Fig. 5j). As shown in high-resolution F1s spectra, the peaks assigned to LiF (~685 eV) in BFE show similar intensities throughout the depth profiling, indicating uniform SEI in BFE electrolyte, while those in DME and DEE exhibit large variation with sputtering. This discussion part has been added to the main text (marked by red color). The corresponding data is added as Fig. 5 and Supplementary Fig. 30.

Note, the data is added as Fig. 5 in the main text.

Note, the data is added as Supplementary Fig. 30.

Reviewer 2:

The manuscript titled “A monofluoride electrolyte with high ionic conductivity for fast-charging and low-temperature lithium metal batteries” presented a new fluorinated electrolyte combining light weight, low viscosity, oxidative stability, and high ionic conductivity. Owing to the monofluoride molecular design, the BFE electrolyte can achieve the high ionic conductivity under wide-temperature conditions, Li metal cycling efficiency (99.75 %), and oxidation stability (4.7 V vs. Li/Li⁺). The BFE-based electrolyte demonstrated excellent cycling performance of pouch cell (areal capacity: 4 mAh cm⁻², N/P: 2, and electrolyte: 2.4 g Ah⁻¹) with energy density (426 Wh kg⁻¹) and high-capacity retention (>80%) during 200 cycles. The full cell with BFE electrolyte

also exhibits stable low-temperature (-30 °C) operation over 150 cycles. However, systematic characterization for the newly proposed high-voltage ether-based solvent is still needed, such as Al corrosion, XPS depth profiling studies of SEI and CEI layers, flammability, long-cycling symmetric Li-Li cell performance and EIS, which is very important for the interface study and realistic Li metal full cell. Overall, I would suggest reconsider after major revision. All the comments are shown as below.

Response: We appreciate the reviewer's valuable and positive comments for our manuscript. The followings are the details of the responses. According to the reviewer's suggestion, systematic characterization including Al corrosion, Cryo-TEM, XPS depth profiling studies of SEI and CEI layers, flammability, stability of symmetric Li-Li cell and interfacial impedance were systematically conducted, and the related detailed discussion for electrochemical results has been explained and discussed in the revised manuscript. We believe that all the requirements for improving our manuscript have been satisfied.

1. The XPS depth profiling studies can reveal the SEI and CEI composition and distribution along the thickness. Can the authors run XPS analysis of the cycled Li-NCM811 full cells with different electrolytes to study the SEI and CEI?

Response: Thank you for your valuable suggestions. The morphology and interfacial components of Li metal anode and NCM811 cathode after cycling are well studied using a scanning electron microscope (SEM), X-ray photoelectron spectroscopy (XPS), and Cryo transmission electron microscope (Cryo-TEM). Here, 5 mAh cm⁻² of Li metal is electrodeposited onto bare Cu foil to examine the deposition morphology and solid electrolyte interphase (SEI) components of Li metals in various electrolytes, including BFE, DME, and DEE. The scanning electron microscopy (SEM) image reveals smooth and compact deposit morphologies of Li metal in BFE electrolyte (Fig. 5a). The compact packing and large crystal size of Li deposits are beneficial to the improvement of cycling stability by reducing the formation of thick SEI and dead Li. However, large amounts of porous Li deposits in both DME and DEE electrolytes easily induce continuous SEI and dendritic growth (Fig. 5b and 5c). At the same plating capacity, the plated Li metal layer in the BFE electrolyte is much thinner than DME and DEE electrolytes (Fig. 5d-5f). The SEI composition of Li metal anode after 50 plating/stripping cycles in BFE electrolyte was also analyzed by low-dose Cryo transmission electron microscopy (Cryo-TEM) (Fig. 5g-5i). A dense and uniform SEI layer with a small thickness of 18 nm is observed in Cryo-TEM images. The inverse fast Fourier transform (IFFT) characterization displays the magnified lattice spacing of 0.20 nm is well-matched with the (200) crystal plane of LiF (Fig. 5h) and the lattice

spacing of 0.27 nm is accorded with the (111) crystal planes of Li_2O (Fig. 5i). This result is also confirmed by the corresponding electron energy loss spectroscopy (EELS) spectra (Supplementary Fig. 30). In addition, X-ray photoelectron spectroscopy (XPS) with etching depth profiles were also used to analyze the SEI components of Li metal in various electrolytes (Fig. 5j). As shown in high-resolution F1s spectra, the peaks assigned to LiF (~685 eV) in BFE show similar intensities throughout the depth profiling, indicating uniform SEI in BFE electrolyte, while those in DME and DEE exhibit large variation with sputtering.

In addition to the stable SEI of Li metal anode, the monofluoride electrolyte also promotes the formation of ultrathin LiF-rich cathode electrolyte interphase (CEI) on high-voltage NCM811 cathode. The CEI components on NCM811 cathodes after 50 charge/discharge cycles are also characterized through high-resolution TEM analysis. As observed in Fig. 5k, a uniform CEI layer with an ultrathin thickness of <1 nm is formed on the surface of the cycled cathode in the BFE electrolyte. Due to the excessive decomposition of the solvents and anions, much thicker CEI layers are generated onto the NCM811 cathode in DME and DEE electrolytes, respectively (Fig. 5l and 5m). In addition, the high-resolution C1s and F1s spectra of the cycled cathode indicate the CEI of BFE electrolyte has more LiF inorganics and fewer organic components than DME and DEE electrolytes (Fig. 5n). Consequently, the ultrathin LiF-rich SEI and CEI significantly contribute to the outstanding cycling stability of Li metal batteries with BFE electrolyte. The discussion parts are added to the revised manuscript (marked in red color). The data are added in Fig. 5 and Supplementary Fig. 30.

Note, the data is added as Fig. 5.

Note, the data is added as Supplementary Fig. 30.

2. Flammability is one of the critical safety issues for LMBs. After decreasing the fluorine amount of the solvent, how about the flammability of the BFE solvent? Please compare the flammability of all fluorinated molecules.

Response: According to the reviewer's suggestion, the flammability of all fluorinated molecules is measured and provided in the supplementary information. As shown in Supplementary Fig. 2, DEE is a highly flammable solvent with a long self-extinguishing time of over 120s. However, when introducing the fluorine atoms, the flammability is significantly restrained by showing a considerably decreased self-extinguishing time for BFE (89s), DFE (72s), TFEE (53s), DFE (54s), and BTFE (22s). It is worth noting that the ethers with higher fluorination offer a short self-extinguishing time and thus better battery safety. Therefore, our BFE electrolyte can improve the safety of the batteries when compared with nonfluorinated electrolytes. The data is added as Supplementary Fig. 2.

Note, the data is added as Supplementary Fig. 2.

3. SEI on Li anode should be characterized by Cryo-EM and also FFT to reveal the SEI composition. The CEI in full cells after cycling also need be analyzed by HR-TEM to check the phase transition or surface lattice reconstruction on NCM811 particles.

Response: Thank you for your valuable suggestions. The morphology and interfacial components of Li metal anode and NCM811 cathode after cycling are well studied using a scanning electron microscope (SEM), X-ray photoelectron spectroscopy (XPS), and Cryo transmission electron microscope (Cryo-TEM). Please check the detailed data and discussion parts in the first question.

4. For the average CE by Aurbach method, can the authors provide reproduced data (three more) for the BFE (99.75%)? I do not expect the reproduced cells to be as high

as 99.75% but some values >99.65% are definitely needed to solidify this outstanding performance.

Response: According to the reviewer's suggestion, we have reproduced the data of the average CE of Li metal in BFE electrolyte by Aurbach's method. As shown in Supplementary Fig. 18, we performed three more tests of the Li metal efficiency using Li||Cu cells, and the data are 99.75%, 99.72%, and 99.64%, respectively, suggesting the excellent stability of our electrolyte to lithium metal, especially during repeated plating/stripping processes. The data is added as Supplementary Fig. 18.

Note, the data is added as Supplementary Fig. 18.

5. CC-CV charge method is widely used in the real battery. Can the authors run the Li-NCM811 full cells through a constant-current-constant-voltage protocol at 4.4V (Nature Energy 7, 94–106 (2022)) at 0.3 C to check the long-cycling performance using BFE electrolyte?

Response: According to the reviewer's suggestion, we have checked the cycling stability of the Li||NCM811 cells through a constant-current-constant-voltage protocol at 0.3 C. As shown in Supplementary Fig. 27, the Li||NCM811 full cell with high areal loading over 3.5 mAh cm⁻² delivers a high reversible capacity of over 200 mAh g⁻¹ when charging to 4.4 V, and can keep high capacity retention over 93% after 100 charge/discharge cycles, suggesting the excellent cycling stability of the BFE electrolyte even testing using a CC-CV method. The corresponding discussion part has been added to the revised manuscript. The data is added as Supplementary Fig. 27.

Note, the data is added as Supplementary Fig. 27.

6. Li||Li symmetric cells are widely used to evaluate interfacial stability. Long-cycling of Li||Li symmetric cells is required for realistic batteries. Can the authors run the Li||Li symmetric cells at 1 mA cm⁻², 1 mAh cm⁻² for long-cycling until cell is failed?

Response: According to the reviewer's suggestion, we have evaluated the interfacial stability of Li metal using Li||Li symmetric cells with a cycling capacity of 1 mAh cm⁻² at 1 mA cm⁻². As shown in Supplementary Fig. 21, the Li||Li symmetric cell in BFE electrolyte shows the smallest overpotential during 500 h plating/stripping processes, which is more stable than that in DME and DEE electrolytes. The discussion part has been added to the revised manuscript and the related data is added as Supplementary Fig. 21.

Note, the data is added as Supplementary Fig. 21.

7. LSV in Li||Al cell is not enough convinced in real battery due to the Al surface passivation and low surface area, can the authors provide LSV results using Super-P-PVDF and Pt electrodes? And the Li||Al cell holds for several hours with increasing voltage (Nature Energy 5, 526–533 (2020))?

Response: According to the reviewer's suggestion, we have checked the oxidation stability of BFE solvent using the Li||Pt cell. As depicted in Supplementary Figure 5c, the oxidation potential of DME is about 4.8 V while the DEE is about 5.0V. However, the real oxidation potential of BFE is increased to 5.2 V due to the fluorination, and this further convinced the excellent stability of BFE molecules at high voltages. To further characterize the stability of different electrolytes to Al current collectors in the real application, we also hold the Li||Al cells for one hour with increasing voltages according to your advice. As shown in Supplementary Figure 5b, when holding the cell at 4.6 V for one hour using DME electrolyte, a large current leakage is observed, while a similar phenomenon is observed for the cell using DEE electrolyte at 4.8 V. In sharp contrast, a much smaller and constant current is observed for the Li||Al cell from 4.0 V to 4.8V using BFE electrolyte, suggesting the excellent Al passivation ability. The data is added in Supplementary Figure 5.

Note, the data is added in Supplementary Fig. 5.

8. Al corrosion caused by FSI anion is serious in the real battery. Can the authors using Li|Al half cell hold at 5 V for 48 hours with different electrolyte and check the Al foil by SEM?

Response: According to the reviewer's suggestion, we have held the Li|Al half cells at 5 V for 48 hours and checked the Al foil by SEM. As shown in Supplementary Fig. 6, porous structure and severe cracks are observed for Al foils in DME and DEE electrolytes, suggesting a typical corrosion behavior by the FSI⁻ anions. In sharp contrast, the Al foils in BFE electrolyte exhibit a uniform and smooth structure, suggesting the excellent Al passivation ability which is consistent with the LSV results. The discussion part has been added to the revised manuscript. The data is added as Supplementary Fig. 6

Note, the data is added as Supplementary Fig. 6.

9. Did the authors try the 3M LiFSI-BFE? Were the various performance better than the 2M LiFSI-BFE?

Response: According to the reviewer's suggestion, we have carried out the cycling test of Li||NCM811 full cells with high mass loading of 3.5 mAh cm⁻² and a small N/P ratio of 1.1 using BFE-2M and BFE-3M electrolytes at 0.3 C. As shown in Supplementary Fig. 26, the reversible capacity of the Li||NCM811 cell using BFE-3M is slightly lower than that of BFE-2M, and the cycling stability is also worse than that of BFE-3M. A possible explanation for this behavior is the increased viscosity and the lower ionic conductivity when increasing lithium salt concentration. The data is added in Supplementary Fig. 26.

Note, the data is added in Supplementary Fig. 26.

10. EIS is essential for the interface study in the LMBs, can the authors conduct the systematic EIS measurement and analysis for the different electrolytes?

Response: According to the reviewer's suggestion, we have tested the impedance of Li||Li cells after cycling for 500 hours. As shown in Supplementary Fig. 22, the impedance of Li||Li cells in BFE (35 ohms) is much lower than that of DME (85 ohms) and DEE (150 ohms) electrolytes, suggesting that better interfacial stability of Li metal anode with less dendrite growth. The data is added in Supplementary Fig. 22.

Note, the data is added in Supplementary Fig. 22.

11. Author mentioned BFE can coordinate Li⁺ ions with one Li-O and two Li-F interaction. Can author grow single crystal and check and diffraction to prove the unique coordination with BFE? Follow the reference Nature Energy 5, 526-533 (2020).

Response: According to the reviewer's suggestion, we have tried to grow a single crystal using lithium triflate ($\text{CF}_3\text{SO}_3\text{Li}$) in BFE solvent according to the reference Nature Energy 5, 526–533 (2020). Unfortunately, we failed to get the single crystal. The coordination between Li^+ ions and BFE molecules has been studied clearly with ^{19}F NMR in Fig. 2, in which we have enough experimental evidence to confirm that Li^+ ions with one Li-O and two Li-F interaction. Besides, the $\text{Li}^+(\text{BFE})$ solvent-separated ion pairs (SSIPs) were also confirmed by the ESI-MS result which shows an apparent peak at the molecule weight of 117 (Supplementary Fig. 16), while no SSIPs peak was observed for DEE electrolyte.

Fig. 2. b-d, ^{19}F NMR of BFE, DFE, and TFFE before and after the salt dissolution.

Supplementary Fig. 16. Electrospray ionization mass spectrometry (ESI-MS) of different electrolytes.

12. BFE solvents exhibited the high CE of 99.75%. Can author measure the performance of anode-free full cell to compare with other literatures?

Response: According to the reviewer's suggestion, we have measured the cycling

performance of the anode-free cells using a commercial loading NCM811 (3.5 mAh cm^{-2}) with our electrolyte. As shown in Supplementary Fig. 28, the Cu||NCM811 cell shows excellent cycling stability and keeps high capacity retention of over 88% after 40 cycles, confirming the outstanding compatibility of BFE electrolyte to Li metal. The data is added in Supplementary Fig. 28.

Note, the data is added in Supplementary Fig. 28.

Reviewer 3:

This work reports the design, synthesis, and application of a novel monofluoride ether, i.e., bis(2-fluoroethyl) ethers, as an electrolyte solvent for high-energy and high-power lithium metal batteries with wide operative temperature ranges. The electrochemical performance is very impressive, particularly at low temperatures. Some experiments and simulations regarding the coordination structure and Li^+ transport have been conducted to interpret the remarkable performance. The results and findings in this work are certainly interesting and important. However, a series of concerns do emerge when the reviewer goes through the article. Moreover, the reviewer has a feeling that the work is not well finished, as some important aspects relevant to the good performance have not been investigated. Therefore, an appropriate decision can only be given after the re-valuation of this article upon a major revision.

Response: We appreciate the reviewer's valuable and positive comments for our manuscript. The followings are the details of the responses. According to the reviewer's suggestion, systematic characterization including Raman spectra, ion transference number, average Coulombic efficiency, Cryo-TEM, and XPS depth profiles of interfacial components of Li metal anode and high-voltage cathode are systematically investigated, and the related detailed discussion for electrochemical results has been explained and discussed in the revised manuscript. We believe that all the requirements for improving our manuscript have been satisfied.

1. The first concern is the cost. It is claimed that the bis(2-fluoroethyl) ethers (BFE) is synthesized with only two basic, low-cost reagents, i.e., 2-fluoroethanol (CAS: 371-62-0) and 1-bromo-2-fluoro-ethane (CAS: 762-49-2). But this is not true. The prices of 2-fluoroethanol and 1-bromo-2-fluoro-ethane are 166 euro/25 g and 160 euro/25 g, respectively (prices from VWR). The authors mentioned that these chemicals were purchased from Aldrich Sigma, but the reviewer did not find them from the web shop. Although the electrolyte leads to the very good performance even under practical conditions, the high cost of the solvent clearly hinders the applications.

Response: According to the reviewer's suggestion, we have checked the price of two reagents (CAS: 371-62-0 and 762-49-2) from chemical reagent companies (Aladdin Reagent and Energy Chemical Reagent). The retail price of 2-fluoroethanol (34 euro/25 g) and 1-bromo-2-fluoro-ethane (14 euro/ 25 g) is much lower than that of VWR. The chemical suppliers claim that the price can be largely reduced if larger amounts of chemicals are required. As such, the synthesis cost of the solvents will not limit its practical application.

2. Why the oxygen from DEE shows lower electron density than the oxygen from DFE and BFE (Figure 2a)? Due to the electron withdrawing effect, the oxygen from DFE and BFE should exhibit lower electron density than that from DEE. Should not be like this?

Response: Thank you for your comments. Due to the electron-withdrawing effect of fluorinated groups, the electron density from oxygen in BFE is indeed lower than that of DEE (BFE-1) in the linear molecular configuration. However, the linear configuration of the BFE molecule can be transformed into a stable five-member-ring molecular structure after coordinating with Li^+ ions (BFE-2). The electron density of oxygen in BFE can be significantly enhanced by sharing the delocalized electrons of monofluoride atoms. This phenomenon also indirectly confirms the interaction of Li-F in BFE electrolytes. The data is added as Supplementary Fig. 7.

Note, the data is added as Supplementary Fig. 7.

3. The authors provide enough results proving the coordination between Li and F from CH_2F and demonstrating the effect of fluoride content on the Li^+ -F coordination. But the experimental characterization on the coordination of Li^+ to FSI and to oxygen from ether is weak or even not mentioned. In the latter MD simulation parts, the authors

mentioned that the aforementioned, not well-characterized coordination also has effects on the Li^+ transport. Therefore, this reviewer recommends the authors to conduct Raman spectra to prove the statements in the MD simulation, and to show the influence of the different fluorinated group on the coordination of Li^+ to FSI⁻ and ether oxygen.

Response: According to the reviewer's suggestion, the Raman and NMR spectra of BFE electrolytes are provided in the supplementary information. As shown in Supplementary Fig. 10, the peak at 820 cm^{-1} is assigned to the ether group in BFE molecules. The peak is shifted to 832 cm^{-1} after the salt dissolution, suggesting the coordination interaction between oxygen and Li^+ ions. The Li-O interaction of BFE electrolyte is also confirmed by the ^{17}O NMR spectrum in Supplementary Fig. 9. In addition, the peak of the CH_2F group at 870 cm^{-1} is shifted to 873.5 cm^{-1} , suggesting the coordination between fluorine and Li^+ ions, which is consistent with the result from the ^{19}F NMR in Fig. 2b. The data is added as Supplementary Fig. 10.

Note, the data is added as Supplementary Fig. 10.

4. How the samples were prepared for NMR measurements, how the chemical shift was calibrated, and what reference was used for the calibration are not mentioned in the experimental section. This leads to the concern on the reliability of some of the results, e.g., some ^{19}F NMR spectra (Fig. 2b, and S5), ^7Li NMR spectra (Fig. 2e) and ^{17}O NMR spectra (Fig. S6), as only single peak appears in each spectrum.

Response: According to the reviewer's suggestion, the detailed NMR measurements of BFE electrolytes are added in the revised manuscript. The NMR samples were prepared and tested using coaxial nuclear magnetic tubes with CDCl_3 , and all the samples were calibrated and locked with the CDCl_3 solvent. All the NMR measurements were performed with a decoupling mode.

5. Fig. 2f shows the solvating energy. How about the contribution from Li⁺-F and Li⁺-O? From the MD simulation (Fig. 3e), the coordination of Li-O(BFE) is clearly stronger or more dominant than Li-FBFE, while the characterization mainly focuses on the coordination between Li⁺ and F from the solvents. What is the respective role of Li⁺-O solvent and Li⁺-F solvent in the proposed solution structure and enhanced Li⁺ transport?

Response: Thank you for your comments. As depicted in Fig. 3a-d, with the monofluoride substitution, the dissociation ability of BFE to LFSI is considerably enhanced compared with that of DEE, which makes a uniform and well-distributed solvation structures of the BFE-2M electrolyte, while highly aggregated ion pairs with inadequately dissociated lithium salts were observed in DEE electrolyte at both room and low temperatures. In addition, from the analysis in Supplementary Fig. 16, we know that the average coordination number for Li-F_{BFE} is 3.4 while the corresponding data of Li-O_{BFE} is 2, also suggesting the significant role of Li-F coordination. The synergetic coordination interaction of Li-F and Li-O contributes to the well-distributed solvation morphologies, and thus enhanced the ionic conductivity up to 8 mS cm⁻¹ and the improved MSD compared with that of DEE electrolyte (1 mS cm⁻¹). From the radial distribution functions in Fig. 3e, the coordination strength of Li-OBFE is indeed stronger than that of Li-F_{BFE}, and this coordination strength is also stronger than that of Li-O_{DEE} (Li-O_{BFE}: 1.98, Li-O_{DEE}: 2.06). It is worth noting that the enhanced coordination ability is ascribed to the monofluoride substitution and the synergetic coordination effect of -CH₂F groups, which has been confirmed by the increased electron cloud density around oxygen in the dominated configuration of BFE molecules (Fig. 2a, and Supplementary Fig. 7). Therefore, the monofluoride substitution and its coordination to Li⁺ ions plays the key role to the big change of BFE electrolyte in physicochemical properties.

The solvation clusters in BFE electrolyte were extracted from the MD simulation trajectories. As shown in Supplementary Fig. 14a, the clusters of two BFE molecules coordinating one Li⁺ ion with strong Li-F and Li-O interaction are observed, which belongs to typical SSIPs and occupy approximately 60% of total Li⁺ ions (calculated according to the coordination number). This SSIPs were also confirmed by electrospray ionization mass spectrometry (ESI-MS) which shows a strong [solvent-Li⁺] mass peak at the mass of 117 in Supplementary Fig. 16. Other clusters consisting of one Li⁺ ions, two BFE molecules, and one FSI⁻ anion are also observed in our electrolyte (Supplementary Fig. 14b-c), which belongs to the CIPs and occupy about 40% of total Li⁺ ions. An interesting phenomenon is that once the FSI⁻ anion participates in the synergistic coordination of the Li⁺ ion, one fluorine is one of the BFE molecules is repulsed away from the Li⁺ ion center, keeping the total coordination number of 6. This

phenomenon strongly indicates that the coordination strength of Li-F_{BFE} is similar to that of Li-O_{FSI}.

Fig. 3 | Investigation of solvation structures via molecular dynamic simulations. a-d, Molecular dynamic (MD) simulation trajectories of DEE (a and b) and BFE (c and d) electrolytes at room and low temperatures, respectively. Li⁺ ions, coordinated solvents, and FSI⁻ anions (within 2.5 Å) are depicted in a ball-and-stick model while the free solvents and FSI⁻ anions are colored semitransparency. Colors for different elements: H-white, Li-lime, C-cyan, N-blue, O-red, F-pink, S-yellow. e, Radial distribution functions comparison of Li-O (BFE), Li-F (BFE), and Li-O (FSI) pairs for DEE and BEE electrolytes at low and room temperatures. f, MSD of Li⁺ ions in BFE and DEE at low and room temperatures.

Supplementary Fig. 14. Solvation clusters extracted from the MD simulation snapshot of the 2 M LiFSI/BFE electrolyte. **a**, solvent separated ion pairs, occupy about 60% of total Li^+ ions, **b-c**, different structures of contact ion pairs.

Note, the data is added as Supplementary Fig. 15.

6. The Li^+ transference number was measured but lead to some concerns. In Fig. S2, the impedance before and after the polarization is very different, and the current does not finally reach a steady state, which affect the reliability of the obtained results. In Fig. S3, the impedance of the cell employing DEE is five times of that with BFE. But the current of DEE cell is still much higher than that of BFE cell, specifically, 6 times. This is clearly wrong. In fact, the impedance from the interphase is more than 100 times of that from the electrolyte at $-30\text{ }^\circ\text{C}$, the adopted electrochemical method is not suitable to get reliable Li^+ transference number.

Response: According to the reviewer's suggestion, the Li^+ transference number of the DEE electrolyte is retested at $-30\text{ }^\circ\text{C}$ by extending the steady time. As shown in Supplementary Figure 4c, the measured current reaches a steady state, and the Li^+

transference number is determined to be 0.49. The updated data has been added as supplementary Fig. 4c.

Note, the data is added as Supplementary Fig. 4.

7. The rate capability and low temperature performance not only relates to the Li^+ transport in the electrolyte but also the interphases. However, the characterization of the interphase is not provided. Therefore, XPS characterizations on the SEI on Li metal should be provided to demonstrate the influence of the fluorinated groups.

Response: According to the reviewer's suggestion, the deposit morphology and SEI composition of Li metal anode after cycling were well studied using scanning electron microscopy (SEM), X-ray photoelectron spectroscopy (XPS), and Cryo transmission electron microscopy (Cryo-TEM). Here, 5 mAh cm^{-2} of Li metal is electrodeposited onto bare Cu foil to examine the deposition morphology and solid electrolyte interphase (SEI) components of Li metals in various electrolytes, including BFE, DME, and DEE. The scanning electron microscopy (SEM) image reveals smooth and compact deposit morphologies of Li metal in BFE electrolyte (Fig. 5a). The compact packing and large crystal size of Li deposits are beneficial to the improvement of cycling stability by reducing the formation of thick SEI and dead Li. However, large amounts of porous Li deposits in both DME and DEE electrolytes easily induce continuous SEI and dendritic growth (Fig. 5b and 5c). At the same plating capacity, the plated Li metal layer in the BFE electrolyte is much thinner than DME and DEE electrolytes (Fig. 5d-5f). The SEI composition of Li metal anode after repeated plating/stripping cycles in BFE electrolyte was also analyzed by low-dose Cryo transmission electron microscopy (Cryo-TEM)

(Fig. 5g-5i). A dense and uniform SEI layer with a small thickness of 20 nm is observed in Cryo-TEM images. The inverse fast Fourier transform (IFFT) characterization displays the magnified lattice spacing of 0.20 nm is well-matched with the (200) crystal plane of LiF (Fig. 5h) and the lattice spacing of 0.27 nm is accorded with the (111) crystal planes of Li₂O (Fig. 5i). This result is also confirmed by the corresponding electron energy loss spectroscopy (EELS) spectra (Supplementary Fig. 30). In addition, X-ray photoelectron spectroscopy (XPS) with etching depth profiles were also used to analyze the SEI components of Li metal in various electrolytes (Fig. 5j). As shown in high-resolution F1s spectra, both BFE and DEE electrolytes exhibit dominant SEI components of inorganic LiF from the anion decomposition. In comparison, large amounts of organic C-F components are observed in DME electrolytes, which is attributed to the side reaction of the DME solvents. This discussion part has been added to the main text (marked by red color). The corresponding data is added as Fig. 5 and Supplementary Fig. 30.

Note, the data is added as Fig. 5.

Note, the data is added as Supplementary Fig. 30.

8. Apart from the very good rate capability, the author reported super high columbic efficiency up to 99.75% at room-temperature and 99.5% at -30 °C. These values refresh the records under the same protocol. Three duplicated cells are recommended to demonstrate the reproducibility.

Response: According to the reviewer's suggestion, we have reproduced the data of the average CE of Li metal in BFE electrolyte by Aurbach's method. As shown in Supplementary Fig. 18, we performed three more tests of the Li metal efficiency using Li||Cu cells, and the data are 99.75%, 99.72%, and 99.64%, respectively, suggesting the excellent stability of our electrolyte to lithium metal, especially during repeated plating/stripping processes. In addition, we also tested another two Li||Cu cells at -30 °C, which also shows similar CEs of 99.38% and 99.46% (Supplementary Fig. 31). The data are added as Supplementary Fig. 18 and 31.

Note, the data is added as Supplementary Fig. 18.

Note, the data is added as Supplementary Fig. 31.

9. Compared to DEE and DME-based electrolytes, BFE-based electrolyte brings to very good cyclability of Li||NMC811 cells, which clearly relates to a more stable cathode/electrolyte interphase on NMC811. Nonetheless, this is not characterized in the manuscript. To make the work more completed, XPS characterizations on the cathode-electrolyte interphase should be conducted.

Response: Thank you for your valuable suggestions. The CEI components on NCM811 cathodes after 50 charge/discharge cycles are also characterized through high-resolution TEM analysis. As observed in Fig. 5k, a uniform CEI layer with an ultrathin thickness of <1 nm is formed on the surface of the cycled cathode in the BFE electrolyte. Due to the excessive decomposition of the solvents and anions, much thicker CEI layers are generated onto the NCM811 cathode in DME and DEE electrolytes, respectively (Fig. 5l and 5m). In addition, the high-resolution C1s and F1s spectra of the cycled cathode indicate the CEI of BFE electrolyte has more LiF inorganics and fewer organic components than DME and DEE electrolytes (Fig. 5n). Consequently, the ultrathin LiF-rich SEI and CEI significantly contribute to the outstanding cycling stability of Li metal batteries with BFE electrolyte. The discussion parts are added to the revised manuscript (marked in red color). The data are added as Fig. 5 and Supplementary Fig. 30.

Note, the data is added as Fig. 5.

REVIEWER COMMENTS

Reviewer #1 (Remarks to the Author):

The authors have well addressed the comments in the previous round so that I would recommend to accept the manuscript as is.

Reviewer #2 (Remarks to the Author):

In the revised version, most of major concerns are addressed. But there still some concerns need be verified by authors before recommendation.

1. Compared to DME, BFE shows lower ionic conductivity, which should give higher polarization. Why did the BFE cell show even lower voltage than DME sample in Supplementary Fig. 21? It seems the DME sample also showed stable polarization during 500 hours.
2. Author said 'As depicted in Supplementary Figure 5c, the oxidation potential of DME is about 4.8 V while the DEE is about 5.0V', which means the DME electrolyte can used to pair with high Ni cathode. But in the literatures, it is known that the DME starts to decompose and is unstable after 4 V. The Li|Pt cell with DME electrolyte in Supplementary Fig. 5 even showed the oxidation stability about 5 V.
3. For the Al corrosion experiment, there is negligible difference between BFE and DEE samples. The low magnification images and pristine Al foil should be provided. In order to compare the anti-corrosion, XPS of Al foil should be added.
4. For the NMR spectra in Fig.2, which reference did author use for F-NMR calibration? It should be added in figure caption. The F chemical shift need be compared after calibration to make sense.

Reviewer #3 (Remarks to the Author):

The authors have addressed some of the issues. But some problems are still there. In addition, the provided new results lead to some new concerns. Further revision is still required before this manuscript can be accepted.

1. For the measurements of Li⁺ transference number, although a final steady current has been reached with the prolonged time, the results are still not reliable. As already mentioned in the previous review report, the impedance from the interphase is more than 100 times of that from the electrolyte at -30 °C, which does not obey the model for the measurement and leads to significant error accordingly to the Bruce-Vincent-Evans equation.

At such the low temperature, the tests of the Li⁺ transference number are very difficult no matter via the Bruce-Vincent-Evans method or NMR. Since the values of the Li⁺ transference number at low temperature are not the core of this work, this reviewer suggests the authors to remove the relevant part from the manuscript, which should be the easiest solution of this problem.

2. XPS measurements have been conducted for the Li metal deposited on Cu in different electrolytes. However, the further analysis on the obtained results are still required. First, scale bar for the intensity is missing, so it is difficult to compare the contents of the specific species in different samples. Second, only the spectra of F 1s are provided, which are not enough to support the statement that LiF is the dominant SEI component. The spectra and fitting analysis of C 1s, N 1s, O 1s, and S 2p should be provided. Third, for the DME-based sample, the peak around 688 eV should be S-F originating from FSI- or its decomposition rather than C-F, because there is no C-F in the electrolyte. For BFE, both C-F from the solvent and S-F from the anion could contribute to the peak at 688 eV, but C-F usually exhibits a higher binding energy. Maybe the analysis of C 1s and S 2p could provide some information to clarify its origin.

3. The experimental details of TEM, SEM, Cryo-TEM and XPS measurements, including the instruments and the sample preparation, are not provided. For XPS measurements, whether the spectra are calibrated, and the etch depth for the sputtering, e.g., nm/min, should be provided also. The composition of the NMC811 electrodes should be provided.

4. Some concerns on the CE tests at -30 °C. The reproduced curves shown in Figure S31 are different from the one shown in Figure 6a. After 20 h, the polarization shown in Figure 6a is much lower than that in Figure S31. In Figure 6a, the polarization of the cell employing BFE after 20 h is similar to that before 20 h, while in Figure S31, the former is apparently larger than the latter. In Figure S31, the overpotential is clearly higher than 15 mV as claimed in the main text. Since the formation cycle was conducted at 30 °C, this information should be marked in Figure 6a and given in the experimental section.

5. XPS measurements of NMC cathodes cycled in different electrolytes have been conducted, leading to some concerns. Usually, the NMC811 cathode not only contains the active material, i.e., NMC811, but also carbon black as conductive additives and PVdF as binder. The peaks from carbon black and PVdF

binder should be observed in C 1s and F 1s spectra, e.g., C-C and C-F, unless the surface of the electrodes is covered by thick CEIs. In this work, the CEIs formed in all the samples are rather thin, ~4nm, according to the TEM images Figure 5k-i. In this situation, the peaks in C 1s and F 1s spectra cannot be assigned to the components of CEIs only, particularly for BFE as its CEI has a thickness lower than 1 nm. Moreover, the peak in the F 1s spectra (Figure 5n) for DME exhibits lower binding energy than the other two curves. Besides, the scale bar for intensity should be provided particularly if the authors want to compare the different samples. In general, more detailed fitting analysis should be conducted for C 1s and F 1s. The analysis of N 1s and S 2p should be equally helpful.

Point-to-point Reply to reviewers' Comments

Reviewer 1:

The authors have well addressed the comments in the previous round so that I would recommend to accept the manuscript as is.

Response: We appreciate the reviewer's very positive and constructive comments for our manuscript.

Reviewer 2:

In the revised version, most of major concerns are addressed. But there still some concerns need be verified by authors before recommendation.

1. Compared to DME, BFE shows lower ionic conductivity, which should give higher polarization. Why did the BFE cell show even lower voltage than DME sample in Supplementary Fig. 21? It seems the DME sample also showed stable polarization during 500 hours.

Response: Thank you for the reviewer's comments. The BFE electrolyte has slightly low ionic conductivities in comparison with the DME electrolyte. However, the Li^+ transference number (~ 0.7 , Supplementary Figure 3) of BFE electrolyte is much higher than that of DME electrolyte (~ 0.3 , Nature Energy 2020, 5, 526-533). The low Li^+ transference number of DME electrolytes easily induces more anions gathering at the surface of Li metal anode during plating/stripping processes, resulting in higher concentration polarization.

2. Author said 'As depicted in Supplementary Figure 5c, the oxidation potential of DME is about 4.8 V while the DEE is about 5.0V', which means the DME electrolyte can used to pair with high Ni cathode. But in the literatures, it is known that the DME starts to decompose and is unstable after 4 V. The Li|Pt cell with DME electrolyte in Supplementary Fig. 5 even showed the oxidation stability about 5 V.

Response: Thank you for the reviewer's comments. According to the LSV result, the oxidation potential of the DME electrolyte is measured to be about 4.8 V and 3.8 V in the Li|Pt, and Li|Al cells, respectively (Supplementary Figure 4a and 4b). The instability of the DME electrolyte in the Li|Al cell can be ascribed to the corrosion effect of electrolyte (e.g., FSI^- anions) on Al foil. As such, the DME electrolyte will decompose at about 3.8 V after pairing with Al foil-supported high Ni cathode, which

is consistent with previously reported works (Nature Energy 2020, 5, 526-533; Nat. Commun. 2015, 6, 6362).

3. For the Al corrosion experiment, there is negligible difference between BFE and DEE samples. The low magnification images and pristine Al foil should be provided. In order to compare the anti-corrosion, XPS of Al foil should be added.

Response: Thank you for the reviewer's comments. The low and high-resolution SEM images of Al foils with various electrolytes were examined after holding at 4.8 V for 5 hours. The Al foil in the BFE electrolyte shows smooth surface morphology, while serious corrosion and cracks are observed in DME and DEE electrolytes, respectively. In addition, the XPS analysis also reveals corrosion-resistant LiF/AlF₃ components are formed at the surface of Al foils in the BFE electrolyte (Supplementary Fig. 6). This result is also accorded with previously reported work (Nature Energy 2020, 5, 291-298). The related discussion parts have been marked in red colors in the revised manuscript.

Supplementary Figure 5. Al corrosion test in different electrolytes via SEM after holding at 4.8 V for 5 hours.

Supplementary Figure 6. Al 2p (a) and F 1s (b) spectra of Al foil after holding at 4.8 V for 5 hours.

4. For the NMR spectra in Fig.2, which reference did author use for F-NMR calibration? It should be added in figure caption. The F chemical shift need be compared after calibration to make sense.

Response: Thank you for the reviewer's comments. The trifluoroacetic acid (CF_3COOH) has been selected as the reference sample for the calibration of fluorinated electrolytes by using coaxial nuclear magnetic tubes. All the NMR measurements were performed with a decoupling mode. The related parts have been added to the figure caption in the revised manuscript.

Reviewer 3:

The authors have addressed some of the issues. But some problems are still there. In addition, the provided new results lead to some new concerns. Further revision is still required before this manuscript can be accepted.

1. For the measurements of Li^+ transference number, although a final steady current has been reached with the prolonged time, the results are still not reliable. As already mentioned in the previous review report, the impedance from the interphase is more than 100 times of that from the electrolyte at $-30\text{ }^\circ\text{C}$, which does not obey the model for the measurement and leads to significant error accordingly to the Bruce-Vincent-Evans equation. At such the low temperature, the tests of the Li^+ transference number are very difficult no matter via the Bruce-Vincent-Evans method or NMR. Since the values of the Li^+ transference number at low temperature are not the core of this work, this

reviewer suggests the authors to remove the relevant part from the manuscript, which should be the easiest solution of this problem.

Response: Thank you for the reviewer's constructive comments. According to the reviewer's suggestion, the related measurements of Li^+ transference number at low temperatures have been removed from the manuscript.

2. XPS measurements have been conducted for the Li metal deposited on Cu in different electrolytes. However, the further analysis on the obtained results are still required. First, scale bar for the intensity is missing, so it is difficult to compare the contents of the specific species in different samples. Second, only the spectra of F 1s are provided, which are not enough to support the statement that LiF is the dominant SEI component. The spectra and fitting analysis of C 1s, N 1s, O 1s, and S 2p should be provided. Third, for the DME-based sample, the peak around 688 eV should be S-F originating from FSI^- or its decomposition rather than C-F, because there is no C-F in the electrolyte. For BFE, both C-F from the solvent and S-F from the anion could contribute to the peak at 688 eV, but C-F usually exhibits higher binding energy. Maybe the analysis of C 1s and S 2p could provide some information to clarify its origin.

Response: Thank you for the reviewer's comments. We have retested the SEIs of Li metal via XPS. The F1s XPS spectra were fitted and analyzed in detail while the scale bars were also added to each spectrum. In addition, the C 1s, O 1s, S 2p, and N 1s spectra (Supplementary Fig. 32-34) were also provided with detailed fitting curves. As shown in high-resolution F 1s spectra, the cycled Li metal in the BFE electrolyte shows uniform LiF SEI components (~ 685 eV) in the whole etching depth, while the SEI component in DME and DEE electrolyte gradually decreases. The other peak around 688 eV is ascribed to S-F components related to the anion decomposition. In addition, the C1s and O1s spectra show that the external SEI of Li metal in BFE electrolyte is mainly organic, while the internal SEI is dominated by inorganic LiF components (Supplementary Fig. 31 and Supplementary Fig. 32). It is worth noting that small amounts of LiN_xO_y and Li_2S_x are also recognized in S2p and N1s spectra for the cycled Li metal with BFE electrolyte (Supplementary Fig. 33 and Supplementary Fig. 34). The related discussion parts have been marked by red colors in the revised manuscript.

Fig. 5. j, F1 s XPS spectra with etching depth profiles of the cycled Li metals in different electrolytes (BFE, DME and DEE).

Supplementary Figure 31. C 1s XPS spectra with etching depth profiles of the cycled Li metals in different electrolytes (BFE, DME and DEE).

Supplementary Figure 32. O 1s XPS spectra with etching depth profiles of the cycled Li metals in different electrolytes (BFE, DME and DEE).

Supplementary Figure 33. S 2p XPS spectra with etching depth profiles of the cycled

Li metals in different electrolytes (BFE, DME and DEE).

Supplementary Figure 34. N 1s XPS spectra with etching depth profiles of the cycled Li metals in different electrolytes (BFE, DME and DEE).

3. The experimental details of TEM, SEM, Cryo-TEM and XPS measurements, including the instruments and the sample preparation, are not provided. For XPS measurements, whether the spectra are calibrated, and the etch depth for the sputtering, e.g., nm/min, should be provided also. The composition of the NMC811 electrodes should be provided.

Response: Thank you for the reviewer's comments. We have added the experimental details of TEM, SEM, Cryo-TEM, and XPS measurements in the revised manuscript and marked them in red color. In addition, the detailed composition of NMC811 electrodes (97 wt% active materials, 1 wt% Super P, and 2 wt% PVDF binder) has also been added to the revised manuscript and marked by red color.

4. Some concerns on the CE tests at $-30\text{ }^{\circ}\text{C}$. The reproduced curves shown in Figure S31 are different from the one shown in Figure 6a. After 20 h, the polarization shown in Figure 6a is much lower than that in Figure S31. In Figure 6a, the polarization of the cell employing BFE after 20 h is similar to that before 20 h, while in Figure S31, the former is apparently larger than the latter. In Figure S31, the overpotential is clearly higher than 15 mV as claimed in the main text. Since the formation cycle was conducted

at 30 °C, this information should be marked in Figure 6a and given in the experimental section.

Response: Thank you for the reviewer's comments. According to your suggestion, we have retested the CE of Li metals at low temperatures as Figure 7a and Supplementary Figure 36 in the revised manuscript (marked by red color).

Fig. 7. a, Li plating/stripping CEs evaluated by Aurbach's measurement at low temperatures of -30 °C.

Supplementary Figure 36. Li plating/stripping CEs tested repeatedly by Aurbach's measurement at low temperatures of -30 °C.

5. XPS measurements of NMC cathodes cycled in different electrolytes have been conducted, leading to some concerns. Usually, the NMC811 cathode not only contains the active material, i.e., NMC811, but also carbon black as conductive additives and PVdF as binder. The peaks from carbon black and PVdF binder should be observed in C 1s and F 1s spectra, e.g., C-C and C-F, unless the surface of the electrodes is covered by thick CEIs. In this work, the CEIs formed in all the samples are rather thin, 4nm, according to the TEM images Figure 5k-i. In this situation, the peaks in C 1s and F 1s

spectra cannot be assigned to the components of CEIs only, particularly for BFE as its CEI has a thickness lower than 1 nm. Moreover, the peak in the F 1s spectra (Figure 5n) for DME exhibits lower binding energy than the other two curves. Besides, the scale bar for intensity should be provided particularly if the authors want to compare the different samples. In general, more detailed fitting analysis should be conducted for C 1s and F 1s. The analysis of N 1s and S 2p should be equally helpful.

Response: Thank you for the reviewer's comments. According to your suggestion, we have performed the XPS measurement and analyzed the C 1s, F 1s, N 1s, and S 2p spectra in detail. As depicted in Fig. 6e, C 1s and F 1s spectra of the cycled cathode indicate the CEI of BFE electrolyte has more LiF inorganics and fewer organic components than DME and DEE electrolytes, which was also accorded with the results of N 1s and S 2p spectra (Supplementary Fig. 35). The related discussion parts have been marked by red colors in the revised manuscript.

Fig. 6. d, e, C1s and F1s XPS spectra of the cycled NCM811 cathodes in different electrolytes (BFE, DME and DEE).

Supplementary Figure 35. S_{2p} (a) and N_{1s} (b) XPS spectra of the cycled NCM811 cathodes in different electrolytes (BFE, DME and DEE).

REVIEWER COMMENTS

Reviewer #2 (Remarks to the Author):

The authors addressed additional comments in the revised manuscript. Now I recommend it to be published.

Reviewer #3 (Remarks to the Author):

The authors have addressed most of my concerns. Just the fitting of XPS results needs to be further revised, particularly for the profile of DEE-based electrolyte in Fig. 6d. Also, please check the fitting in Fig. S32, 33, and 34. Some of the fitted curves do not match with the experimental results.

After the authors revised these minor problems, this manuscript is ready for publication.

Point-to-point Reply to reviewers' Comments

Reviewer 2:

The authors addressed additional comments in the revised manuscript. Now I recommend it to be published.

Response: We appreciate the reviewer's very positive comments for our manuscript.

Reviewer 3:

The authors have addressed most of my concerns. Just the fitting of XPS results needs to be further revised, particularly for the profile of DEE-based electrolyte in Fig. 6d. Also, please check the fitting in Fig. S32, 33, and 34. Some of the fitted curves do not match with the experimental results. After the authors revised these minor problems, this manuscript is ready for publication.

Response: Thank you for the reviewer's comments. We has updated the XPS measurement and fitting of the cycled NCM811 cathode in DEE electrolyte. In addition, we have also revised the fitting curves of Fig. S32, 33, and 34, which are accorded with the Cryo-TEM results. The internal SEI of cycled Li metal in BFE electrolyte is dominated by inorganic LiF and Li₂O components (Fig. 6e and Supplementary Fig. 32). A small amount of LiN_xO_y and Li₂S_x components are also recognized for the cycled Li metal with BFE electrolyte in the revised S2p and N1s XPS spectra (Supplementary Fig. 33 and Supplementary Fig. 34). The related discussion parts have been marked in red colors in the revised manuscript.

Fig. 6. d, C1s and F1s (e) XPS spectra of the cycled NCM811 cathode in different electrolytes (BFE, DME and DEE).

Supplementary Figure 32. O1s XPS spectra with etching depth profiles of the cycled Li metals in different electrolytes (BFE, DME and DEE).

Supplementary Figure 33. S_{2p} XPS spectra with etching depth profiles of the cycled Li metals in different electrolytes (BFE, DME and DEE).

Supplementary Figure 34. N_{1s} XPS spectra with etching depth profiles of the cycled Li metals in different electrolytes (BFE, DME and DEE).

Reviewer #3 (Remarks to the Author):

The authors addressed additional comments in the revised manuscript. Now I recommend it to be published.

Point-to-point Reply to reviewers' Comments

Reviewer 3:

The authors addressed additional comments in the revised manuscript. Now I recommend it to be published. The authors addressed additional comments in the revised manuscript. Now I recommend it to be published.

Response: We appreciate the reviewer's very positive and constructive comments for our manuscript.